

# Freshwater resources under success and failure of the Paris climate agreement

Jens Heinke[1], Christoph Müller[1], Mats Lannerstad[2], Dieter Gerten[1,3], Wolfgang Lucht[1,3]

[1]Potsdam Institute for Climate Impact Research, PO Box 60 12 03, 14412 Potsdam, Germany
[2]Independent Consultant, Welanders väg 7, SE-112 50 Stockholm, Sweden
[3]Humboldt-Universität zu Berlin, Department of Geography, Unter den Linden 6, 10099 Berlin, Germany

*Correspondence to*: Jens Heinke (heinke@pik-potsdam.de)

**Abstract.** Population growth will in many regions increase the pressure on water resources and likely increase the number of people affected by water scarcity. In parallel, global warming causes hydrological changes which regionally also impact human water supply. This study estimates the increase in pressure on global water resources due to population growth and adverse hydrological effects at different levels of global mean temperature rise above pre-industrial level ($\Delta T_{glob}$), including reduced mean water availability, growing prevalence of hydrological droughts, and increased frequency of flooding hazards. The study analyses the results in the context of success and failure of implementing the Paris Agreement, and evaluates how climate mitigation can reduce the future number of people affected by severe hydrological change, assessed for the population as a whole, as well as for vulnerable population groups already projected to experience water scarcity in the absence of climate change. The results show that without climate mitigation efforts, in 2100 more than 5.1 billion people in the SSP2 population scenario would more likely than not be affected by severe hydrological change, and about 1.9 billion of them would already be affected by water scarcity in the absence of climate change. Limiting warming to 2 °C or 1.5 °C by a successful implementation of the Paris Agreement would strongly limit the number of people affected by severe hydrological changes and water scarcity to 274 million or 104 million, respectively. At the regional scale, substantial water related risks remain at 2 °C, with more than 10 % of the population affected in Latin America and the Middle East and North Africa region. Constraining $\Delta T_{glob}$ to 1.5 °C would limit this share to about 5 % in these regions.

## 1 Introduction

Within the 2030 Agenda for sustainable development of the United Nations (United Nations 2015), 'access to clean water and sanitation' is one of the 17 Sustainable Development Goals (SDGs). For other SDGs, such as 'zero hunger' and 'affordable and clean energy', access to sufficient water resources is a precondition (International Council for Science 2017). Already today, more than 2 billion people live in countries where total freshwater withdrawals exceed 25 % of the total renewable freshwater resource (United Nations 2017). Population increase and economic development are expected to further increase pressure on water resources leading to enormous challenges for water resource management to maintain or

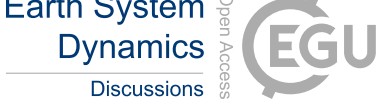



increase water supply. Climate change potentially aggravates this challenge in some regions by changing the hydrologic conditions to the worse. Such changes include a reduction in total physical water availability, but also a change in the flow regime, which may lead to more frequent or more severe drought events or an increased risk in flooding (Döll and Schmied 2012). All these changes affect the water supply management and will make meeting the demand and achieving SDGs more

costly or impossible.

As of April 2017, 194 countries responsible for >99 % of global greenhouse gas emissions have signed the Paris climate agreement that aims at "holding the increase in the global average temperature to well below 2 °C above pre-industrial levels and to pursue efforts to limit the temperature increase to 1.5 °C above pre-industrial levels" (UNFCCC 2015). However, the Intended Nationally Determined Contributions (INDCs) submitted by countries so far are insufficient to achieve this goal,

probably leading to a median warming of 2.2 to 3.5 °C by 2100 if no further efforts will be taken (Rogelj et al. 2016). With the announced withdrawal of the US from the agreement and all major industrialized countries currently failing to meet their pledges (Victor et al. 2017), even a more extreme warming cannot be ruled out. It is therefore timely to assess the climate change impacts associated with a success (limit warming to 1.5 or 2 °C) and a failure of the Paris Agreement (exceeding 2 °C). The purpose of this study is to provide such an assessment for the water sector by systematically quantifying changes in

renewable freshwater resources at different levels of global warming between 1.5 and 5 °C above pre-industrial levels in steps of 0.5 °C. The most extreme level of 5 °C thereby marks an upper boundary consistent with the median warming for a scenario without climate policy (3.1–4.8 °C) (Rogelj et al. 2016).

Most global assessments of climate change impacts on water resources have employed a measure of water stress like the water crowding index (WCI; Falkenmark 1989) or the water scarcity index (WSI; World Meteorological Organization 1997).

A main difficulty in the interpretation of studies based on the WCI or WSI is that results are presented in classes ("stressed" vs. "non-stressed") and changes within these classes are not analysed, although a change from "stressed" to "even more stressed" may be much more relevant than small changes across the classification borders. Another shortcoming of both indices is that they rely on mean annual water availability and do not capture seasonal shortages and changes in variability. In order to gain a detailed and comprehensive understanding of changes in the water sector, this study directly analyses

climate impacts with respect to decrease in mean water availability, growing prevalence of hydrological droughts, and increase of flooding hazards. To estimate these hydrological changes, three key metrics are used to assess flow regime changes: (i) mean annual discharge (MAD); (ii) the number of drought months (ND); and (iii) the 10-year flood peak (Q10). By combining these changes with spatially explicit population projections consistent with Shared Socioeconomic Pathways (SSPs) (Jones and O'Neill 2016), the number of people exposed to severe hydrologic changes is estimated for each level of

$\Delta T_{glob}$.

However, looking at the total number of affected people provides only limited insights about the societal risks of hydrological changes. These risks are greatly determined by the underlying population-driven water scarcity level – that is, in already stressed regions a certain change in hydrologic conditions is probably more significant than the same change in unstressed regions. To account for this aspect and to facilitate a more detailed interpretation, we apply the WCI to estimate

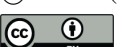



the future population pressure on water resources under the assumption of no climate change and analyse hydrological changes within each WCI stress class. A specific focus is on the exposure to severe levels of change in the most vulnerable water scarcity classes, where hydrological changes will matter the most and management challenges are expected to be particularly high.

**2 Methods**

**2.1 Climate scenarios**

In order to systematically assess climate change impacts on freshwater resources, we use the PanClim climate scenarios described in Heinke et al. (2013). The dataset consists of 8 different scenarios of $\Delta T_{glob}$ obtained with the MAGICC6 model (Meinshausen, Raper, and Wigley 2011) based on greenhouse gas emissions that result in a range of warming levels above

pre-industrial (~1850) conditions from 1.5 °C to 5.0 °C in steps of 0.5 °C in 2100 (2086–2115 average). For each $\Delta T_{glob}$ pathway, the local response in climate variables is emulated for 19 different General Circulation Models (GCMs) from the Coupled Model Intercomparison Project phase 3 (CMIP3) ensemble using a pattern-scaling approach. In so doing, normalized climate anomalies (changes per 1.0 °C of $\Delta T_{glob}$ increase) of temperature, precipitation and cloud cover for each month of the year in each 0.5 arc-degree grid cell are obtained by linear regression between time series of climate variables

and the corresponding time series of $\Delta T_{glob}$. The unexplained variance of these linear models is in the same order (temperature and cloud cover) or only slightly larger (precipitation) than inter-annual variability in the pre-industrial control run without anthropogenic forcing, indicating that most of the climate change information is captured by the obtained patterns. The normalized climate anomalies are used to calculate local climate anomalies for any given $\Delta T_{glob}$ relative to the year 2009 (when $\Delta T_{glob}$ was 0.9 °C above pre-industrial level). These local climate anomalies were then applied to monthly

reference time series of local climate that represent average conditions and variability in 2009.

A total of 152 climate scenarios (8 $\Delta T_{glob}$ pathways x 19 GCM patterns) for the period 1901–2115 are obtained. Up to the year 2009, time series are based on precipitation and cloud cover from CRU TS3.1 (Harris et al. 2014) and precipitation from the GPCC full reanalysis dataset version 5 (Schneider et al. 2014). The reference time series for the period 2010–2115 to which climate anomalies are applied is created from the historic datasets by resampling. Further details on the climate

scenarios have been described by Heinke et al. (2013).

**2.2 Impact model**

For assessing the impacts of climate change on the hydrological cycle, we employ the LPJmL Dynamic Global Vegetation Model version 4 (LPJmL4) that simulates the growth of natural vegetation and managed land in coupling with the global carbon and hydrological cycle (Schaphoff, von Bloh, et al. 2017). The model has been extensively evaluated showing good

performance in representing the global hydrological cycle (Rost et al. 2008; Schaphoff, Forkel, et al. 2017). LPJmL has been



widely applied in water resources assessments (D. Gerten et al. 2011; Rockström et al. 2014; Steffen et al. 2015; Jägermeyr et al. 2016).

For the simulations conducted here, the model is first run without land use for a spinup period of ~5000 years using pre-industrial atmospheric $CO_2$ concentrations and climate data from 1901–1930. This is followed by a second spinup of 390 years up to 2009, during which atmospheric $CO_2$ concentrations and climate vary according to historical observations, and constant land use of the year 2000 is prescribed (Fader et al. 2010). All 152 scenario simulations are initialized from this state, assuming constant land use over the whole simulation period and atmospheric $CO_2$ concentrations consistent with the respective $\Delta T_{glob}$ scenario (Heinke et al. 2013). All simulations are performed without direct anthropogenic intervention on freshwater resources (water withdrawals and dam operation) as their effect are assumed to be captured by the WCI.

In addition to the 152 $\Delta T_{glob}$ scenarios one additional simulation for the period 2010–2115 is carried out using the reference climate data without any anomalies applied and with constant atmospheric $CO_2$ concentrations of the year 2009. This simulation represents a no climate change setting, for which transient time series with inter-annual variability but without a general trend are produced. This scenario is used as the reference simulation for the comparison with the other climate scenarios. Because the sequence of dry and wet years is identical in all scenarios and the reference case, any differences between the scenarios and the no-climate-change reference simulation can be attributed to global warming.

## 2.3 Hydrological change metrics

The focus of this study is on hydrological changes due to climate change that are relevant from a water resource perspective. With "water resources" we refer to 'blue' water—the water that can be withdrawn from rivers, lakes and aquifers, and which can be directly managed by humans—as opposed to 'green' water, i.e. the soil moisture in the root zone from local precipitation that can only be used by locally growing plants (Rockström et al. 2014).

We here use river discharge as an approximation of the blue water resource. The river discharge of a grid cell simulated by LPJmL includes all the water that enters the cell from upstream areas and all surface and subsurface runoff generated within the cell, calculated at daily time step. Although water is often withdrawn from lakes and aquifers, no more than the possible recharge to these storages can be withdrawn over a prolonged period. Therefore, river discharge as computed with LPJmL represents a good approximation of the total renewable blue water resource (excluding non-renewable fossil groundwater from aquifers with very long recharge times).

Three metrics relevant from a water resource perspective, i.e. mean annual discharge (MAD), the number of drought months (ND), and the 10-year flood peak (Q10), are calculated for each grid cell for each level of $\Delta T_{glob}$ and each GCM pattern. Based on these results we determine in each grid cell the lowest level of $\Delta T_{glob}$ at which levels defined as severe change in each indicator are transgressed in more than 50 % of GCM runs (at least 10 out of 19). This corresponds to the 'more likely than not' category used in IPCC AR5 (Mastrandrea et al. 2011).



### 2.3.1 Mean water availability

Changes in mean annual discharge (MAD) are used as a measure for changes in mean water availability, assuming that a substantial decline in MAD undermines the capacity of existing water supply infrastructure to continuously fulfil societal water demands. Depending on how much of MAD is already appropriated, investment in improved infrastructure can help to maintain stable supply. If options for supply management are exhausted or become too costly, demand management options (increasing end-use efficiency and allocative efficiency) need to be considered (Ohlsson and Turton 1998). We define a decrease in MAD by 20 % or more as a severe change that requires some form of management intervention (either on the supply or the demand side). The same threshold was also used by Schewe et al. (2014) to define severe decrease in annual discharge.

### 2.3.2 Hydrological drought

The occurrence of prolonged periods of below-average discharge, mostly initiated by interannual climate variability, is referred to as hydrological drought. To provide stable water supply to society, water supply systems are adjusted to seasonal variability and drought regimes. A substantial increase in drought periods thus impairs the capability of existing water management infrastructure.

We apply a drought identification method proposed by Van Huijgevoort et al. (2012) to determine which months of a monthly time series of river discharge are in drought condition. The method is based on a combination of the threshold level method (TLM) and the consecutive dry month method (CDM). The TLM method classifies a month as drought-stricken if it falls below a given threshold (here, the month-specific discharge that is exceeded 80 % of the time). However, in ephemeral rivers a method that accounts for the duration of dry periods is more appropriate since the TLM would classify all months with zero flow as drought. We adopt this combination of TLM and CDM from Van Huijgevoort et al. (2012) but make some modifications to obtain a more robust and plausible algorithm. First, a month-specific discharge threshold is applied to identify drought months according to the TLM method. If the TLM threshold is zero and thus the number of drought months in a given calendar month exceeds 20 %, the CDM is used to determine which of the months with zero discharge can be classified as drought months. To this end, the number of preceding consecutive TLM droughts is determined for each month with zero discharge in the given calendar month. Then, a threshold is selected that retains only the months with the longest preceding dry period so that the total number of drought months in that calendar month is 20 %. By applying the TLM and CDM thresholds estimated for the no climate change simulation to the discharge time series from simulations under different climate scenarios the change in the number of drought months (ND) can be estimated. We define an increase in the total number of drought months by 50 % (i.e., from 20 % to 30 %) as a severe change that will require an upgrade of existing water management systems to maintain a reliable water supply.



### 2.3.3 Flood hazard

All water supply infrastructure should be designed to withstand typical flooding events. A flood with a return time of the 50-100 years (Q100) is typically used as a reference case (Coles 2001) but spillways of critical infrastructure such as dams and reservoirs are designed for flood events with a return time of 1000 years or more (Dyck and Peschke 1995). An increase in

the magnitude of these design floods poses a serious threat to water management systems with potentially disastrous consequences.

The magnitude of extreme events with long return periods can be derived from much shorter observed time series of annual maximum floods by fitting a suitable extreme value distribution (e.g. a Gumbel or Generalized Extreme Value distribution; Coles 2001). The obtained extreme value distribution is then used to extrapolate the magnitude of flood events with long

return periods. Because this procedure is computationally expensive and introduces additional uncertainties, we here analyse changes in the annual maximum flood with a return period of 10 years (Q10). The magnitude of this event is directly derived from the simulated time series of monthly discharge by determining the maximum annual flood that is exceeded in 3 out of 30 years (technically a return time of 10.33 years).

We define an increase in Q10 by 30 % as severe change that needs to be addressed by investment in enhancing flood

resistance of water supply infrastructure or by changing reservoir operation schemes to increase the safety buffer for flood protection (at the cost of storage capacity for water supply). Although a threshold of 30 % appears rather high, it accounts for the fact that changes in Q10 are only a proxy for changes in more extreme flood events. Due to the shape of typical extreme value distributions, the increase in magnitude of flood events with longer return time will be lower that the change in Q10.

### 2.4 Grid-based water crowding indicator

In order to determine where transgressions of severe change levels in the three metrics matter most, we estimate which part of global population is experiencing water stress in the absence of additional climate change. We use the WCI originally proposed by Falkenmark (1989) to assess different levels of population on water resources from population change. Originally, the water crowding index has been designed to be applied at country scale, which may hide important within-country variations (Arnell 2004). Therefore and because the calculation procedure is more straightforward, it has become

more common to calculate it at basin scale (Falkenmark and Lannerstad 2004; Dieter Gerten et al. 2013; Arnell and Lloyd-Hughes 2014; Gosling and Arnell 2016). In this paper, we develop a new calculation procedure to obtain a measure of water crowding that can be calculated and interpreted at grid-cell scale. This can then be combined with the simulated hydrological changes at grid-cell scale to estimate hydrological change for different levels of water crowding.

The simplest way for calculating the grid cell water crowding is by relating total discharge (equivalent to the sum of all

upstream and local runoff) to the sum of upstream and local population. Although probably appropriate in many cases, this can lead to an overestimation of crowding (pressure on available water) if a substantial proportion of runoff is generated in parts of the basin with low population. In order to calculate the effective population pressure on the total available water





within each grid cell, we therefore treat local (within grid cell) runoff and the inflow from each upstream cell $i$ separately. While local runoff $w_0$ is assumed to be fully available to the local population $p_0$, the inflow from each upstream cell $w_i$ is equally shared between local population $p_0$ and effective upstream population $p'_i$ (eq. 3) corresponding to that inflow:

$$w' = w_0 + \sum_{i=1}^{N} w_i \cdot \frac{p_0}{p'_i + p_0} \qquad (1)$$

5 The obtained effective water quantity $w'$ is the effective available water in that grid cell. Relating local population $p_0$ to $w'$ yields the effective water crowding index $WCI'$:

$$WCI' = \frac{p_0}{w'} \qquad (2)$$

for the respective cell. Multiplying $WCI'$ with the total water $w$ (sum of local runoff and all inflows) gives the effective population $p'$ that is required for the calculation of $WCI'$ in the downstream cell:

$$p' = WCI' \cdot w = p_0 \frac{w}{w'} \qquad (3)$$

Because $p'$ of all upstream cells must be known to determine $WCI'$, the calculation for a whole basin starts at the fringes (in cells with no inflow, i.e. where $w_i = p'_i = 0$) and continues consecutively to the basin outlet.

Different levels of $WCI'$ represent different levels of population pressure on water resources. In the original publication of the water crowding indicator (Falkenmark 1989), crowding conditions below 100 people per flow unit (p/fu; 15 1 fu = 1e6 m$^3$ per year) are considered as *well-watered*, with *moderate problems* occurring between 100 and 600 p/fu, and *water stress* occurring between 600 and 1000 p/fu. Beyond 1000 p/fu population experiences *chronic water scarcity*, and the level of 2000 p/fu is interpreted as a "water barrier" beyond which *acute water scarcity* prevails.

## 3 Results

### 3.1 Change in water crowding driven by population change

20 Between 1950 and 2010 the number of people that live with acute water scarcity according to the WCI computed at grid cell scale has increased from 108 million (4.3 % of global population) to 888 million (13.0 % of global population) due to population growth alone. At the same time, the number of people that live under chronic water scarcity has increased from 176 million (7.0 % of global population) to 758 million people (11.1 % of global population) (Fig. 1c and 1d).

This trend is projected to continue in the future under all five SSP population scenarios (Fig. 1c and 1d). The total number of 25 people in either scarcity class in 2100 is projected to be higher than today (2010) in all scenarios, with higher global population associated with higher numbers of affected people. The share of population under acute water scarcity is also projected to increase substantially in all scenarios to 16.7–27.2 % in 2100. In contrast, the share of global population affected by chronic water scarcity increases only slightly to 11.5–13.8 % in 2100.

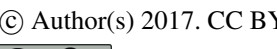



The total number of people living in under well-watered conditions or experiencing only moderate supply problems is projected to remain relatively stable at today's 4.49 billion in most scenarios (4.21–5.05 billion), with a somewhat higher number (6.22 billion) found in the most populated SSP3 scenario. Thus, most of the additional global population in 2100 will experience some degree of water stress or scarcity. This is driven by two developments: (1) population increase in places

with high water crowding, and (2) more places becoming water stressed due to population increase.

### 3.2 Severe changes in hydrologic conditions under different levels of $\Delta T_{glob}$

In many regions, severe changes in MAD, ND and Q10 occur under the majority of climate change patterns within the range of $\Delta T_{glob}$ considered in this study (Fig. 2a-c). Thus, a substantial proportion of global population (shown for SSP2 in 2100, for other populations scenarios see supplementary Fig. S2) will 'more likely than not' be exposed to severe changes in

hydrologic conditions. If $\Delta T_{glob}$ would reach 5 °C in the absence of any climate policy, 1.09 billion people (12.2 % of global population) would be affected by a severe increase in MAD, 1.27 billion people (14.1 %) by a severe increase in droughts, and 1.41 billion people (15.7 %) by a severe increase in Q10. The pace at which these levels are reached with increasing $\Delta T_{glob}$ is not linear and differs for the three metrics. For MAD and ND, the additional number of people that become affected at each step of $\Delta T_{glob}$ first increases and then declines again, with the larges increment occurring between 2 °C and

2.5 °C for MAD and between 1.5 °C and 2 °C for MAD; the increment of people becoming affected by severe changes in Q10 steadily increases with $\Delta T_{glob}$.

If global warming was limited to 2 °C by a successful implementation of the Paris Agreement, the number of people affected by severe hydrological change could be strongly limited, with 108 million people (1.9 %) affected by a severe decrease in MAD, 319 million (5.0 %) by a severe increase in drought months, and 15 million (0.2 %) by a severe increase in floods. If

warming was limited to 1.5 °C these figures could be reduced further by 63 % for MAD, 72 % for droughts, and 79 % for floods. However, even a partial failure of the Paris Agreement with an exceedance of the two-degree target by only 0.5 °C would increase the number of people affected to MAD, ND, and Q10 by 353 million, 235 million, and 77 million, respectively. Thus, ambitious mitigation efforts can substantially reduce the number of people affected by severe changes in MAD and ND, while their effect on the number of people affected by severe changes in flooding remains comparatively low.

In order to synthesize the individual results for the three metrics, we analyse at which level of $T_{glob}$ severe hydrological changes of any type occur for the majority of the climate change patterns (Fig. 2d). The result is not just the sum of the individual patterns of MAD, ND, and Q10 but also includes many additional grid cells where strong but diverging changes are projected by the ensemble of climate models. The total number of people more likely than not affected by severe hydrological change reaches 5.1 billion (57.0 %) at 5 °C warming in the SSP2 population scenario. If warming was

successfully limited to 2 °C, this number would be reduced to 636 million (7.1 %). At 1.5 °C warming, only 199 million (2.2 %) would be affected. If the two-degree target is missed by 0.5 °C (1 °C), 525 million (1121 million) more people would be affected.



### 3.3 Severe changes and water crowding combined

The challenge to adapt water supply systems to severe hydrological changes increases with growing population pressure. Thus, to assess where hydrological changes pose the biggest societal threat, we in combination analyse hydrological change and water crowding. Comprehensive results with the proportion of population in each crowding class that is more likely than

not affected by severe changes in MAD, ND, Q10 or any combination of these three are shown in Fig. S3. For reasons of clarity we present the results for two aggregated classes of water crowding: high water crowding with >1000 p/fu and moderate to low water crowding with ≤1000 p/fu (Fig. 3).

Under the assumption of no climate change, as much as 2.99 billion people (33.3 % of total population) are estimated to experience high water crowding in 2100 in the SSP2 scenario. For all measures of hydrologic change and across the whole

range of $\Delta T_{glob}$, the proportion of people affected by a severe change in the high crowding category is larger than among those experiencing moderate to low crowding (Fig. 3). This asymmetric distribution of impacts is most pronounced for MAD, Q10, and the combined metric (Figs. 3a, 3c, and 3d) and largely independent of the population scenario (Fig. S4). Thus, severe hydrological change is more likely to occur in places where the potential to adapt water supply systems to these changes is low.

Because of the reduced potential to adapt to severe hydrological change, the population under high water crowding is the primary concern when analysing climate change impacts on water resources. At 5 °C warming, 491 million people affected by high crowding (5.5 % of total population) are more likely than not affected by a severe change in MAD, 457 million (5.1 %) by a severe change in droughts, 614 million (6.8 %) by a severe change in floods, and 1.94 billion (21.6 %) by a severe hydrological change of any kind. A successful implementation of the Paris Agreement sufficient to limit warming to

2 °C would dramatically reduce these figures to 102 million (1.1 %) for MAD, 154 million (1.7 %) for droughts, 8.4 million (0.1 %) for floods, and 274 million (3.1 %) for the combined metric.

However, the remaining number of people affected by severe hydrological change as well as the implications of more ambitious mitigation efforts, or a failure of the Paris Agreement, differs greatly among world regions (Table 1, for countries assigned to each region see Fig. S5). About 59 % of the 274 million people that live under high water crowding and are more

likely than not affected by a severe hydrologic change at 2 °C warming, live in Latin America (LAM) and the Middle East and North Africa region (MEA), where they make up more than 11 % of total population. Another 29 % live in South Asia (SAS) and Sub-Saharan Africa (SSA), but due to high total population numbers their share remains below 2 %. The high share of affected population in LAM and MEA is particularly worrying since substantial societal and economic efforts will be needed in these regions to protect these people against the consequences of severe hydrological change and to achieve

water-related SDGs. More ambitious mitigation efforts that keep warming below 1.5 °C would reduce the number of affected people by more than half, to 5.4 % in MEA and 4.2 % in LAM. This would clearly increase the chances and reduce the costs for achieving water-related SDGs in these regions. In all other regions, the share of affected population would drop below 1%.




Failure of the Paris Agreement would substantially increase water related climate risks in many regions. In six out of ten regions, the number of affected people more than doubles if the 2 °C target is exceeded by only 0.5 °C. This is most severe in the MEA region, where almost one quarter of the total population would be affected. But also in SSA and Europe (EUR), the proportion of affected population rises to around 5 %. Between 2.5 °C and 3 °C warming the increases in number of
affected people is strongest in South Asia (SAS), SSA, North America (NAM), and EUR. At 4 °C warming, the share of affected population exceeds 10 % in 7 out of 10 regions, with MEA, Australia-New Zealand, SAS, SSA, and LAM being most strongly affected. At 5 °C warming, the share of affected population reaches almost 40 % in MEA and 30 % in ANZ, exceeds 20 % in SAS, SSA, and LAM, and exceeds 15 % in NAM, EUR, and East Asia (EAS). In Russia and Central Asia (RCA) and Southeast Asia (SEA) the share of affected people remains below 5 %, due to a low share of population under
high water crowding and less severe hydrologic change.

Although numbers differ among population scenarios, the overall pattern of where and how much change occurs in the different regions is consistent across all SSP population scenarios. A comprehensive overview over population under high water crowding and affected by severe hydrologic change in different world regions for all population scenarios is given in Fig. S6.

**4 Discussion**

Our estimate that 24.1 % of global population today live under acute or chronic water scarcity (>1000 p/fu), is well within the range of 21.0–27.5 % (average 24.7 %) reported by previous studies applying the WCI on river basin level (Gerten et al. 2013; Arnell and Lloyd-Hughes 2014; Kummu et al. 2016). Estimates of future SSP population living in river basins with >1000 p/fu under present-day climate conditions are given by Arnell & Lloyd-Hughes (2014), who estimate a range of 39.5–
54.2 % of affected global population across different SSP scenarios. This is considerably higher than the range of our estimates of 29.0–41.0 %, but due to the lack of other comparable studies, it is not clear whether these discrepancies are caused by the choice of the hydrological model or by the difference in scale (basin or grid cell) at which the WCI is calculated. However, using the same model as in our study, Gerten et al. (2013) estimate that 38.5 % of global population in the SRES A2r population scenario would live in river basins with >1000 p/fu under current climate conditions, which is
close to our estimate of 41.0 % for the SSP3 scenario, to which the A2r scenario is comparable in terms of total population (12.3 billion compared to 12.6 billion in 2100). In contrast, the corresponding estimate from by Arnell & Lloyd-Hughes (2014) is as high as 54.2 % for the SSP3 scenario, which indicates that LPJmL generally tends to produce lower estimates of future population affected by water scarcity.

A direct comparison of hydrological changes estimated here to previous studies is not straightforward due to the unique
design of this study. Only few global studies have assessed climate change impacts on water resources as function of $\Delta T_{glob}$ (Gerten et al. 2013; Schewe et al. 2014; Gosling and Arnell 2016), but they typically focus on changes in mean annual discharge and number of people affected by water scarcity. A relevant study for comparison is Schewe et al. (2014), which





analyses changes in MAD obtained from an ensemble of ten global hydrological models (GHMs) forced by climate scenarios from five different GCMs. The overall pattern of changes in MAD simulated by LPJmL across 19 GCMs agrees well with results from Schewe et al. (2014), but exhibits a generally lower magnitude of changes (see Fig. S7 and Fig. 1 in Schewe et al. (2014)). Thus, MAD changes simulated by LPJmL (both increases and decreases) tend to be smaller than

simulated by most other GHMs. This becomes even more apparent when comparing the percentage of people affected by a 20 % decrease in MAD. For a $\Delta T_{glob}$ of 2.5 °C (equivalent to an additional warming of 1.9 °C relative to the control simulation) we estimate a median share of 8.6 % of affected global population across all GCMs. This is substantially less than the median value of 13 % of affect population estimated for 2 °C additional warming by Schewe et al. (2014) and approximately represents their lower end of the interquartile range. This can be attributed to the response of dynamic

vegetation in LPJmL that is not included in most other GHMs (Schewe et al. 2014).

In summary, the global and regional estimates of population affected by high water crowding and severe hydrological change obtained from LPJmL are lower than from most other GHMs. Thus, population affected by water scarcity and severe hydrological change should be regarded as conservative estimates.

**5 Conclusions**

Future water stress will be affected by population growth and climate change, which are both subject to uncertainty and heterogeneous distribution patterns. Population increase alone leads to a disproportionally strong increase in population affected by water scarcity in all 5 SSP population scenarios analysed here, increasing the number of people most vulnerable to hydrological change. In addition, severe hydrological change caused by climate change is unequally distributed among water stress classes and will most strongly affect population under water scarcity, already coping with limited room for

further adaptation.

If global warming would continue unabated to reach 5 °C above pre-industrial levels in 2100, 5.1 billion people (57.0 % of global population) in the SSP2 population scenario would more likely than not be affected by severe hydrological change. 1.9 billion people of them (21.2 % of global population) would have limited capacity to adapt due to high population pressure on water resources. With a successful implementation of the Paris Agreement limiting global warming to 2 °C, the

number of people affected by severe hydrological change could be reduced to 636 million people (7.1 % of global population), of which 274 million (3.1 %) would have limited capacity to adapt. If temperature increase could be limited to 1.5 °C, the number of people exposed to climate-driven water challenges could be further reduced to 199 million (2.2 %) and 104 million (1.2 %), respectively.

Due to the heterogeneous spatial distribution of water scarcity and severe hydrological change, the proportion of affected

population with little capacity to adapt at 2 °C warming still exceeds 10 % in the Latin America and the Middle East and North Africa region. Such a high share of severely affected population bears a substantial risk for failing of achieving water-related SDGs. Thus, 2 °C mean global warming cannot be considered a safe limit of warming in these regions. More



ambitious mitigation efforts that would keep warming at, or below, 1.5 °C would substantially reduce that risk by reducing the share of population affected by severe hydrological change but little capacity to adapt to about 5 %.

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

**Figure 1: Spatial pattern of water crowding in 2010 (a) and in 2100 for SSP2 population (b). Absolute (c) and percentage share of total population (d) in different water crowding classes from 1950 to 2010 and from 2011 to 2100 in five different SSP population scenarios under current water availability, i.e. assuming no climate change.**





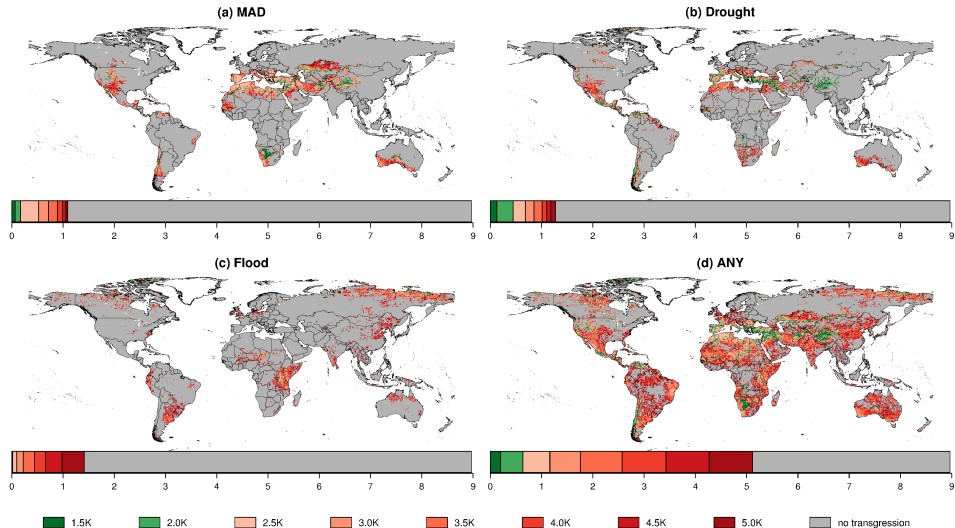

**Figure 2:** $\Delta T_{glob}$ **at which critical hydrological changes occur in more than half of the GCMs (10 out of 19), with (a) mean annual discharge, (b) number of drought months, (c) 10-year flood peak, and (d) any of these. Bars underneath the maps indicate population affected by respective changes for the SSP2 population scenario.**

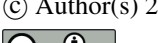



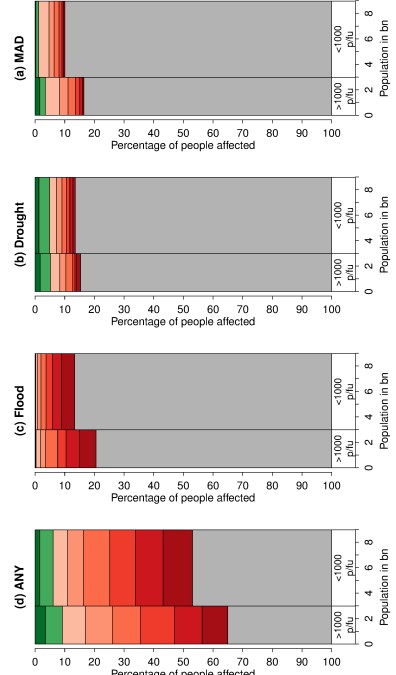

**Figure 3: Figure 3. Proportion of SSP2 population in 2100 in two aggregate water crowding classes, high (>1000 p/fu) and low (≤1000 p/fu), affected by critical change in (a) mean annual discharge, (b) number of drought months, (c) 10-year flood peak, and (d) any of these. Total number of people in each crowding class is given on the y-axis, the proportion of people affected in each class is given on the x-axis. Color scale for $\Delta T_{glob}$ same as in Fig. 2.**



**Table 1: Number of people in 2100 for the SSP2 population scenario affected by high population pressure on water resources (>1000 p/fu) and severe hydrological change in different world regions (percentage of total population per region in brackets). Regions are: MEA (Middle East and North Africa), ANZ (Australia and New Zealand), SAS (South Asia), SSA (Sub-Saharan Africa), LAM (Latin America), NAM (USA and Canada), EUR (Europe, excluding Russia), EAS (East Asia), RCA (Russia and Central Asia), SEA (Southeast Asia).**

| | Total Population | Population > 1000 p/fu | Population with > 1000 p/fu and affected by severe hydrologic change | | | | | |
| --- | --- | --- | --- | --- | --- | --- | --- | --- |
| | | | 1.5 °C | 2.0 °C | 2.5 °C | 3.0 °C | 4.0 °C | 5.0 °C |
| MEA | 740 | 383 | 40.1 | 86.7 | 174.8 | 200.3 | 245.0 | 283.5 |
| | | (51.8%) | (5.4%) | (11.7%) | (23.6%) | (27.1%) | (33.1%) | (38.3%) |
| ANZ | 51 | 26 | 0.0 | 0.6 | 1.5 | 1.5 | 9.6 | 15.4 |
| | | (50.7%) | (0.0%) | (1.2%) | (2.9%) | (2.9%) | (18.8%) | (30.2%) |
| SAS | 2282 | 883 | 16.3 | 34.6 | 74.1 | 147.8 | 417.1 | 587.5 |
| | | (38.7%) | (0.7%) | (1.5%) | (3.2%) | (6.5%) | (18.3%) | (25.7%) |
| SSA | 2395 | 820 | 14.6 | 45.0 | 115.6 | 218.6 | 390.5 | 544.5 |
| | | (34.2%) | (0.6%) | (1.9%) | (4.8%) | (9.1%) | (16.3%) | (22.7%) |
| LAM | 662 | 221 | 27.8 | 74.8 | 82.1 | 100.3 | 129.7 | 153.8 |
| | | (33.3%) | (4.2%) | (11.3%) | (12.4%) | (15.1%) | (19.6%) | (23.2%) |
| NAM | 510 | 142 | 3.9 | 14.4 | 18.8 | 38.4 | 54.8 | 77.0 |
| | | (27.9%) | (0.8%) | (2.8%) | (3.7%) | (7.5%) | (10.8%) | (15.1%) |
| EUR | 579 | 148 | 0.1 | 14.7 | 28.5 | 53.2 | 62.2 | 104.0 |
| | | (25.6%) | (0.0%) | (2.5%) | (4.9%) | (9.2%) | (10.7%) | (18.0%) |
| EAS | 913 | 220 | 0.0 | 0.6 | 4.9 | 12.0 | 80.0 | 142.7 |
| | | (24.1%) | (0.0%) | (0.1%) | (0.5%) | (1.3%) | (8.8%) | (15.6%) |
| RCA | 198 | 40 | 1.4 | 2.5 | 3.5 | 5.7 | 8.1 | 9.6 |
| | | (20.1%) | (0.7%) | (1.3%) | (1.8%) | (2.9%) | (4.1%) | (4.9%) |
| SEA | 642 | 105 | 0.0 | 0.0 | 0.9 | 2.4 | 6.7 | 22.3 |
| | | (16.4%) | (0.0%) | (0.0%) | (0.1%) | (0.4%) | (1.0%) | (3.5%) |
| World | 8971 | 2988 | 104 | 274 | 505 | 780 | 1403 | 1940 |
| | | (33.3%) | (1.2%) | (3.1%) | (5.6%) | (8.7%) | (15.6%) | (21.6%) |

