# Peer review of "Figure S1: Spatial patterns of water crowding in 2010 and for five different population scenarios in 2100 under current water availability, i.e. assuming no climate change."

_Earth System Dynamics, 2017_

## Referee Comment (RC1) · Anonymous Referee #1 · 8 Mar 2018

The following is a review of the article titled, "Freshwater resources under success and failure of the Paris climate agreement" by Heinke et al.

In this study, the authors apply a simplified impact model (LPJmL) under 152 climate scenarios (8 dT x 19 GCMs) and 5 SSP population scenarios to demonstrate the global populations at-risk of severe water crowding, which to varying degree will be exacerbated by projected changes to water availability. The authors define the following three critical thresholds: a greater than 20% decreases in mean annual discharge, an increase of 50% in the total number of drought months, and an increase in the 10-year return flood event by 30%. Under the SSP2 scenario, they report that 33% of the world's population will be living in >1000 p/fu regions and 9.1% of those affected will be in region's with severe hydrologic change, and thus, presumably the most challenging

adaptation cases. The paper is interesting and worthwhile but not carefully written nor illustrated. There are a number of ways in which this paper can be improved.

Although I am not a water manager or decision maker, I do not see that the information provided is particularly actionable. The regions are quite large. Can the authors provide a subset of the most striking examples of sub-regions under population and hydrologic change stress (i.e., map the areas of Table 1 for each degree warming?).

The authors also do not discuss regional uncertainty in the GCMs or SSPs. For example, Fig. S7 shows (white) areas where there is disagreement among models and conversely, agreement among models. I would think at least the model spread (as a surrogate for projection uncertainty) should be discussed in-hand with the water stress projections.

I also misunderstand the statistics for "ANY" hydrologic changes (e.g., Fig 2 and throughout). Take the Indian subcontinent, for example, which is colored in panel (d) but not for (a-c). How can the "ANY" affected area exceed the sum of the area for the 3 metrics of severe change?

The "majority", "more likely than not" and other terminology was introduced but not consistently or clearly applied throughout the paper.

I provide my detailed comments below.

Abstract: the fact that MAD, ND, and Q10 are used to quantify variability needs explicit mention.

Ln9 suggest something closer to the following: "by water scarcity. [Simultaneously, global warming is shifting the seasonality and overall quantity of available water. This study estimates the separate and joint effects of population growth and hydroclimate change on global water resources for a range of warming scenarios, ranging from +1.5-5C. Hydroclimate change is quantified through three metrics: mean annual discharge, number of drought months, and magnitude of the ten-year return flood event. ...and

evaluates how climate [change] mitigation...hydrological change, as well as the severity of the water scarcity. The results show that without climate [change] mitigation..."

Pg2In1-5 there should be discussion of changing precipitation phase (snow vs rain), earlier springs, longer growing seasons, and glacial melt (Himalayas).

Pg2ln23 it is unclear how the current approach addresses "changes in variability"

Ph2ln27 the number of drought months does not tell us about the severity of the drought. And multi-year droughts are not identified because every year is treated independently.

Pg3ln14 suggest [0.5 x 0.5 degree grid cell]

Pg3In22-23 where does temperature come from? In cases for which GPCC and CRU precipitation are both available, which one is used, or how are the two estimates merged? These datasets revert to climatology in months for which observations are not available. So, the inter-annual variability of the random sample will be less than actual unless these years are filtered.

Pg3ln24 suggest "[random] resampling [with replacement]"

Pg3ln27 what are the inputs and parameters for LPJmL?

Pg5In7 it should be noted that in practice the seasonality of the MAD shortfall matters, since the reservoirs (and rivers) are managed for alternative purposes including flood management and hydroelectric power generation

Pg5In16 is this "river discharge" or "grid cell runoff". There is no mention of a routing scheme. Is the CDM approach most needed because the analysis is on a grid-by-grid basis?

Pg5ln26 why not just use the standardized precipitation index-6months (SPI-6)?

Pg6ln14 large floods can be important "drought busters". What is the logic for pe-

СЗ

nalizing Q10 floods in a water scarcity context? Could the authors provide further justification for the 30% threshold? What is the relative frequency shift? i.e., Q10 plus 30% is what return flood on-average for each region?

Pg6ln19 the authors have omitted a section introducing SSP scenarios

Pg7Ins11 what are the "basins" that were used. Can these be illustrated?

Pg8In4-5 it would be useful for the authors to provide figures that better illustrate the relative/shared contribution of both pathways named here.

Pg8ln8 define "substantial"

Pg8ln9 what is "severe"

Pg8Ins17-19 are these statistics from a table or figure or not shown? Is it 108 million fewer or total impacted? 319 million fewer or total? 15 million fewer or total? Unclear if "these figures" are total affected or differences from 5C numbers is prior paragraph.

Pg8Ins16-30 is there no feedback between the population scenario and water availability? Will population continue to grow as projected despite severe shortages of water?

Pg9In2 "water supply systems" statement is vague and should be qualified by references

Pg9In9 where regionally are these 2.99 billion people concentrated?

Pg9In11 where geographically is this change in high crowding projected?

Pg9In13 "occur in places", please provide examples.

Pg9In18 I misunderstand. How is ANY (1.94billion) larger than the sum of MAS, ND, and Q10?

Pg9In21 it would be instructive to see the regional distribution of the projected benefit of the Paris agreement

Pg9In22 intended reference to "remaining number of people affected by severe hydrological change" is unclear.

Pg9In26 clarify here if "11% of total population" is of global affected population (274mil) or LAM or MEA population

Pg9ln26 is this "another 29% [of the affected global population] live in SAS and SSA, although they locally comprise only 2% of the population"?

Pg9In28 "since substantial societal and economic efforts". Please clarify or add a reference.

Pg9ln31-32 "reduce the costs" this may not always hold. Depending on the region and the solution, the infrastructure investment to serve 5% may be just as expensive as serving 10%.

Pg10ln3 "one quarter of the total population" I am a bit confused with this statistic following the 11% figure cited in the prior paragraph.

Pg10In14 a version of Fig S6 should be included in the main article.

Pg11In16 the meaning/intent of "disproportionally strong" is unclear

Pg11In19 suggest "affect populations [already coping with water scarcity]. Since the specific affected populations have not been identified or discussed, I do not believe a statement about "room for further adaptation" can be supported here.

Pg11In22 the phrase "more likely than not" should be italicized and applied more methodically throughout where appropriate.

Pg11In22-23 suggest "hydrological change. [Of those affected, 1.9 billion (21.2% of global population) would have..."

Pg11ln23 "limited capacity to adapt". Again, I do not believe the study supports statements about barrier to adaptation. It is unclear that population pressure [always] limits

capacity to adapt.

Pg11In30 separate statistics should be cited for Latin America and Middle East and North Africa regions

Pg12In1-2 Is this statement sill in reference to LA and MEA or globally, in-general?

Fig1 the flow unit should be defined in the caption. Why is SSP3 so different in terms of total population growth? I'd personally prefer a figure with panels for each region with pop. Growth and % in each water class.

Fig 2. The specific thresholds that define "critical", i.e. 20% decrease in MAD, 50% increase in ND, and 30% increase in Q10 should be noted here.

Fig3. Clarify in the caption that this data corresponds with Fig.2. Why is the population affected by the two classes unvarying? It appears to be 3billion for all cases for <1000 p/fu. Why not just limit the x-axis to 70% for readability? Big takeaways- droughts are impacted at all warming levels, MAD above 2C, and Q10 above 3C?

Table 1. What is the scale factor for the population? The map of the regions delineated should be included in the main paper. Does the 33% of global population correspond to the "any" category in Fig 2d? Does the 21.6% of global population affected at +5C correspond with the "any" category in Fig 3d? Where would the 65% from Fig 3d fall in this table, if there was an additional section? Clarify that all percentages are provided as a percentage of total global population.

Fig S1 Most differences appear in Middle East, China, South Asia, and North Africa. The discussion and supporting figures could do better to highlight this finding.

Fig S2 why does the SSP2 scenario no match what is shown in Fig.3?

Fig S3 "water [crowding]"

Fig S4 "water [crowding]"

Fig5 this belongs in the main article.

Fig S6 consider a version of this in the main article. Perhaps 10 panels covering each region? FlgS7 Why not include this plot and parallel plots for ND and Q10 in the main article?

---

## Referee Comment (RC2) · Anonymous Referee #2 · 3 Apr 2018

The manuscript "Freshwater resources under success and failure of the Paris climate agreement" is within the scope of the journal, novel, generally well written and state of the art.

However, there are some MAJOR issues that need to be addressed before publication:

* It is good that a manuscript is concise. However, in the description of the results I find it not really concise, as in the mean time the description of data used and methodology lacks information. A more detailed description of the latter is needed to better understand the results.

* Climate change projections: What I find particularly missing in the manuscript is the topic of sea level rise. Under all average global temp rise due to climate change,

the sea level will rise. The authors discuss temp rise until 5degrees C, so different sea level rises will occur. Why has this not been accounted for in the manuscript? E.g. coastal flooding will occur without appropriate adaptation. Is this included in the 10-year flooding scenario of the authors? Please discuss. Add a new section under section 4 on issues like this and other uncertainty/limitations of the study.

* The metric MAD, whether or not in combination with the water crowding indicator: The authors write on page 2 lines 22-23 on the important topic of "... seasonal shortages and changes in variability". They quote that it is important to address this. However, by using a metric like MAD, I do not see at all that seasonality or changes in seasonality due to climate change are addressed. Like in mountain regions, winter can have more water availability due to climate change but summer less. In a mean annual metric this is not accounted for. Also in the water crowding indicator this seasonality is not represented. Please discuss, and again in a new section under section 4 "other uncertainty/limitations of the study". A new publication in STOTEN about water stress partly discusses these issues - https://doi.org/10.1016/j.scitotenv.2017.09.056 . Please discuss relating to this paper.

MODERATE/MINOR COMMENTS

* Page 2 Line 4 "the water supply" delete "the"

* Page 2 Lines 32-34: "more significant". Why? I do not see this. Why is this change in water scarcity more significant in already stressed than unstressed regions. Is an increase in water stress not important in any region? Are water users and the environment not affected in both situations? Please justify this statement or alter it. * Page 3 Lines 2-4: same comment

* Population growth: give more information on quantities and assumptions in the scenario's used

* Page 7 Lines 1-17: Does the water crowding-indicator account for

ground water and environmental flows? Please discuss, again referring to https://doi.org/10.1016/j.scitotenv.2017.09.056

* page 8 line 11: increase in MAD, or is it decrease? Discuss in more detail - more precipitation but also ET, so what happens with resulting MAD

---

## Author Comment (AC1) · 29 Jun 2018

We wish to thank the anonymous reviewer for her/his comprehensive and constructive comments. Below, we provide detailed responses and propose changes in the manuscript.

*Comment: Although I am not a water manager or decision maker, I do not see that the information provided is particularly actionable. The regions are quite large. Can the authors provide a subset of the most striking examples of sub-regions under population and hydrologic change stress (i.e., map the areas of Table 1 for each degree warming?).*

[Figure]

Response: The main objective of the paper is to inform the climate mitigation discussion with an emphasis on the implications of success and failure of the Paris climate agreement. The focus is therefore on temperature levels at which transgression of critical thresholds may occur as well as on the number of people affected at different temperature levels. This kind of information may not be particularly actionable for water managers, but it is not the paper's intention to provide that. In this light, the table with regionally affected population merely serves a breakdown of global aggregated affected population presented before, to highlight that: (i) affected population is unevenly distributed around the globe; and (ii) the extent to which numbers of affected population increase with temperature also differs greatly across regions. A further breakdown into smaller sub-regions would bear the risk of obscuring these massages by adding too much detailed information. Also, the second aspect would be difficult to see in a set of maps.

Changes in manuscript: There is certainly potential to sharpen the key messages in the paper (not only those related to Table 1) and to state the objective of the paper more clearly. We will address this in the revised manuscript.

*Comment: The authors also do not discuss regional uncertainty in the GCMs or SSPs. For example, Fig. S7 shows (white) areas where there is disagreement among models and conversely, agreement among models. I would think at least the model spread (as a surrogate for projection uncertainty) should be discussed in-hand with the water stress projections.*

Response: In this paper, GCM uncertainty is addressed by analysing only impacts that are 'more likely than not'. This is also the case for the regional results in Table 1. Analysing model spread represents an alternative way of dealing with projection uncertainty and has been extensively applied in existing literature. We consider the approach taken in this paper to be more suitable with the intended objective to inform the climate mitigation discussion. The model spread shown in Fig. S7 only serves as a basis for comparison of our projections with existing literature, namely Schewe et
al., 2014. The impact of population scenario uncertainty has been taken into account explicitly by analysing results for all five SSP population scenarios. However, since our findings show that most aspects of population exposure to severe hydrological change do not vary much across population scenarios, we have decided to focus on the medium scenario SSP2 in the main paper and show detailed results for all other SSPs in the SI only.

Changes in manuscript: We see that there are a few places in the manuscript where the reference to the SI material can be improved. We will address this in the revised manuscript.

*Comment: I also misunderstand the statistics for "ANY" hydrologic changes (e.g., Fig 2 and throughout). Take the Indian subcontinent, for example, which is colored in panel (d) but not for (a-c). How can the "ANY" affected area exceed the sum of the area for the 3 metrics of severe change?*

Response: Fig. 2d shows the temperature for each pixel, where severe hydrological change of any of the three types occurs in 10 out of 19 GCMs. For example, this could be a combination of 5 GCMs showing transgression in MAD and 6 other GCM showing transgression in ND, which booth would not show up in the individual maps for these metrics.

Changes in manuscript: We will improve the explanation of the Fig. 2d and the ANY metric.

*Comment: The "majority", "more likely than not" and other terminology was introduced but not consistently or clearly applied throughout the paper.*

Changes in manuscript: We will revise manuscript to assure consistent use of terminology.

*Comment: Abstract: the fact that MAD, ND, and Q10 are used to quantify variability*

*needs explicit mention.*

Response: Although we regard the consideration of variability related hydrological change as an important contribution, we do not think it needs to be emphasized in the abstract. Instead, we believe the paper will benefit from a stronger alignment along the term 'severe hydrological change', which summarizes severe changes in aspects that are related to variability (ND and Q10) and aspects that are not (MAD).

Changes in manuscript: Rewrite to assure consistent use of the term 'severe hydrological change' throughout the paper. In particular, we will add a short definition of 'severe hydrological change' in the abstract and a more detailed motivation in the introduction.

*Comment: Ln9 suggest something closer to the following: "by water scarcity. [Simultaneously, global warming is shifting the seasonality and overall quantity of available water. This study estimates the separate and joint effects of population growth and hydroclimate change on global water resources for a range of warming scenarios, ranging from +1.5- 5C. Hydroclimate change is quantified through three metrics: mean annual discharge, number of drought months, and magnitude of the ten-year return flood event. . . .and evaluates how climate [change] mitigation. . .hydrological change, as well as the severity of the water scarcity. The results show that without climate [change] mitigation. . ."*

Response: We much appreciate these improvements to the abstract. However, the main focus of our study is on severe hydrological change related to climate change and not on water crowding/scarcity. The latter only serves the purpose to highlight where severe hydrological change may matter the most and be of societal relevance. We see that the abstract does not make this clear in the best possible way and we will revise the abstract to improve this aspect.

Changes in manuscript: Revise abstract to improve clearness of study objectives and elements.

[Figure]

*Comment: Pg2ln1-5 there should be discussion of changing precipitation phase (snow vs rain), earlier springs, longer growing seasons, and glacial melt (Himalayas).*
Response: We agree that the paper would benefit from an elaboration of what we mean by 'changing the hydrological conditions' here.
Changes in manuscript: Include a sentence/section that lists examples of changing hydrological conditions due to climate change.

*Comment: Pg2ln23 it is unclear how the current approach addresses "changes in variability"*
Response: We agree that "changes in variability" is a very broad term and that this sentence requires refinement to express more clearly what is missing in existing analyses and what will be address in this study.
Changes in manuscript: Rephrase sentence/section.

*Comment: Ph2ln27 the number of drought months does not tell us about the severity of the drought. And multi-year droughts are not identified because every year is treated independently.*
Response: During preparation of the paper we have experimented with various metrics of drought severity but found the outcomes to be very inconsistent and highly dependent on the chosen metric. We therefore decided to use the number of drought months as a metric for changes in droughts, which is simpler and more transparent. ND as calculated here does in fact cover multi-year droughts as the CDM method is applied to the entire continuous time series.
Changes in manuscript: Explicitly mention that ND is able to capture multi-annual droughts.

*Comment: Pg3ln14 suggest [0.5 x 0.5 degree grid cell]*
Changes in manuscript: Apply proposed changes.

*Comment: Pg3ln22-23 where does temperature come from? In cases for which GPCC and CRU precipitation are both available, which one is used, or how are the two estimates merged? These datasets revert to climatology in months for which observations are not available. So, the inter-annual variability of the random sample will be less than actual unless these years are filtered.*

Response: There is a typo in this line, it should read: "[..], time series are based on temperature and cloud cover from CRU TS3.1 [. . .]." For precipitation, only GPCC is used. The problem that datasets revert to climatology in months where observations are missing, is primarily an issue in the first half of the 20th century in the CRU dataset. For the construction of the reference time series the period 1961 – 2009 was used, and it should therefore be largely unaffected by this problem. In addition, the months instances where the climatology was used are difficult to identify and their removal would introduce artifacts in the reference time series (only full years and global fields were resampled to retain the temporal and spatial autocorrelation of climate data). Changes in manuscript: Correct typo.

*Comment: Pg3ln24 suggest "[random] resampling [with replacement]"*
Changes in manuscript: Apply proposed changes.

*Comment: Pg3ln27 what are the inputs and parameters for LPJmL?*
Response: The method section provides detailed information about the climate inputs used to drive LPJmL in this study. A paper including a full model description including all standard input data and parameters has just been published (Schaphoff, von Bloh, et al., 2018), along with a comprehensive model validation paper (Schaphoff, Forkel, et al., 2018). At the time of submission of the present manuscript these papers were still pending final publication. Hence, only the corresponding discussion papers of both these papers could be referenced. These refences will be updated in the revised

version of the present manuscript.
Changes in manuscript: Update references of model documentation and validation.

*Comment: Pg5ln7 it should be noted that in practice the seasonality of the MAD short-fall matters, since the reservoirs (and rivers) are managed for alternative purposes including flood management and hydroelectric power generation*
Response: Decrease in MAD (mean annual discharge over 30 years) is a metric for the decline in long-term average water availability. It lacks any information of seasonality or interannual variation, which is the very reason why we have complemented it with metrics capturing changes in droughts and floods.
Changes in manuscript: None.

*Comment: Pg5ln16 is this "river discharge" or "grid cell runoff". There is no mention of a routing scheme. Is the CDM approach most needed because the analysis is on a grid-by-grid basis?*
Response: It is river discharge. The routing scheme is in not mentioned, indeed. The CDM approach is needed to identify droughts in ephemeral rivers.
Changes in manuscript: Add description of routing scheme.

*Comment: Pg5ln26 why not just use the standardized precipitation index-6months (SPI-6)?*
Response: SPI and related indices are used to detect meteorological drought whereas the focus in this paper is on hydrological drought.
Changes in manuscript: None.

*Comment: Pg6ln14 large floods can be important "drought busters". What is the logic for penalizing Q10 floods in a water scarcity context? Could the authors provide further*

*justification for the 30% threshold? What is the relative frequency shift? i.e., Q10 plus 30% is what return flood on-average for each region?*

Response: Because the return time of large floods (with a typical return time of 100 years or more) is difficult to estimate from a 30-year time series, we use an increase in Q10 as a proxy for an increase in such large floods. For typical extreme value distributions, the increase in Q100 will always be lower than the increase in Q10, but the exact relationship is not known and depends on the exact shape of the extreme value distribution. Therefore, the 30 % threshold is inherently difficult to justify. What can be done, and that is indeed missing here, is to give a range of possible increases in Q100 and Q1000 that correspond to a 30 % increase in Q10 based on a range of typical shapes (parameter combinations) of extreme value distributions. This would then also make clear that the focus of this metric is on large floods rather than Q10 itself.

Changes in manuscript: Rephrase and clarify. Add typical ranges for increases in magnitude (and/or decrease in return time) of Q100 and Q1000 corresponding to a 30% increase in Q10.

*Comment: Pg6ln19 the authors have omitted a section introducing SSP scenarios*

Response: SSP storylines are extensively covered in existing literature precise knowledge about these storylines is not needed for the interpretation of results in this paper. Nevertheless, mentioning at least the scenario names is certainly useful.

Changes in manuscript: Add section with short SSP scenario description.

*Comment: Pg7lns11 what are the "basins" that were used. Can these be illustrated?*

Response: River basins are defined by the drainage network also used for river routing in the model. However, the description of the routing network is missing (see comment above) and will be added in the revision.

Changes in manuscript: Add description of routing network.

*Comment: Pg8ln4-5 it would be useful for the authors to provide figures that better illustrate the relative/shared contribution of both pathways named here.*
Response: The focus of this paper is on population exposure to severe hydrological change due to global warming. The water crowding analysis merely serves as a means to estimate where theses changes may matter the most. We do not think that knowing how much of future water stressed population is caused by each of the two pathways is relevant in this context. In particular, as the amount of information is quite large (5 SSPs times 5 crowding classes). In addition, it may not be so easy to disentangle the two cases as population will often continue to grow after a pixel has tipped into water stress or scarcity.
Changes in manuscript: None.

*Comment: Pg8ln8 define "substantial"*
Response: Another (and probably better) word that would fit here is 'considerable'. While we agree that both terms are rather vague, a more precise wording is not required at this point. Detailed information on population affected by severe hydrological change at different levels of global warming is provided in the remainder of this section.
Changes in manuscript: None.

*Comment: Pg8ln9 what is "severe"*
Response: Thresholds for what consist a severe change in MAD, ND, and Q10 are defined and motivated in the method section. However, we see that it is not entirely clear that these thresholds are referred to here.
Changes in manuscript: Add clarification that 'severe change' relates to the thresholds defined in the methods. Name them explicitly here again.

*Comment: Pg8lns17-19 are these statistics from a table or figure or not shown? Is it 108 million fewer or total impacted? 319 million fewer or total? 15 million fewer or total? Unclear if "these figures" are total affected or differences from 5C numbers is prior paragraph.*
Response: All numbers mentioned in this section are displayed in Fig. 2. All three numbers mentioned by the reviewer are totals not differences. Changes in manuscript: Clarify that these are totals.

*Comment: Pg8lns16-30 is there no feedback between the population scenario and water availability? Will population continue to grow as projected despite severe shortages of water?*
Response: Population scenarios and their spatial disaggregation are given by the SSP storylines. Thus, potential deviations from the SPP projections which 'in reality' would be the result following impacts from water scarcity, are not captured in this study, nor are they in any other such study, to our knowledge. It is worth mentioning that under extreme scarcity population growth could not be sustained or that a shift towards non-conventional water resources (e.g., desalination) may occur. Changes in manuscript: Add a comment on that in methods or discussion.

*Comment: Pg9ln2 "water supply systems" statement is vague and should be qualified by references*
Changes in manuscript: Rephrase and/or qualify with references.

*Comment: Pg9ln9 where regionally are these 2.99 billion people concentrated?*
Response: This can be seen in the first column of Table 1 but is not explicitly mentioned in the text since the focus of the paper is on severe hydrological change and not on water crowding. However, we will revise this section to give the regional results more weight and may include a reference to regional water crowding numbers.

Changes in manuscript: Rephrase/clarify and add reference if required.

*Comment: Pg9ln11 where geographically is this change in high crowding projected?*
Response: This seems to be a misunderstanding: water crowding is a static overlay in our analysis, calculated from river discharge under contemporary climate conditions and future population patterns (here SSP2). We will make sure to make this clearer in the revised manuscript. The regional distribution of population experiencing water scarcity and severe hydrological change is given in Table 1 and discussed in the main text.
Changes in manuscript: Rephrase/clarify.

*Comment: Pg9ln13 "occur in places", please provide examples.*
Response: This not as easy as it seems. Water crowding often occurs in places of small spatial extent (a few grid cells) but high population density (i.e., cities). We do not think that the scale of analysis (global model, pattern scaled climate scenarios, 0.5 degree resolution) allows to report results at the scale of cities. Regional examples are given in Table 1 and are discussed in the text.
Changes in manuscript: None.

*Comment: Pg9ln18 I misunderstand. How is ANY (1.94billion) larger than the sum of MAS, ND, and Q10?*
Response: This is because in the ANY category, population is in at least 10 out of 19 GCM exposed to a severe hydrological of any kind. Thus, all combinations of severe for MAD, ND, and Q10 are possible here. For example, a transgression of MAD in 5 GCMs and a transgression of ND in 6 other GCM, would mean a pixel and its population is considered in the ANY category but neither in the MAD nor the ND category. While we have tried to provide a definition of ANY in the paper (p8ln25-28), we acknowledge that correct understanding is crucial and deserves more space.

Changes in manuscript: Improve explanation of ANY category.

*Comment: Pg9ln21 it would be instructive to see the regional distribution of the projected benefit of the Paris agreement*
Response: This is shown in Table 1 and described in the following section.
Changes in manuscript: None

*Comment: Pg9ln22 intended reference to "remaining number of people affected by severe hydrological change" is unclear.*
Changes in manuscript: Rephrase and clarify.

*Comment: Pg9ln26 clarify here if "11% of total population" is of global affected population (274mil) or LAM or MEA population*
Changes in manuscript: Rephrase and clarify.

*Comment: Pg9ln26 is this "another 29% [of the affected global population] live in SAS and SSA, although they locally comprise only 2% of the population"?*
Response: This is correct.
Changes in manuscript: Apply proposed change.

*Comment: Pg9ln28 "since substantial societal and economic efforts". Please clarify or add a reference.*
Changes in manuscript: Rephrase/clarify and add reference if required.

*Comment: Pg9ln31-32 "reduce the costs" this may not always hold. Depending on the region and the solution, the infrastructure investment to serve 5% may be just as*

*expensive as serving 10%.*

Response: Under conditions of unconstrained water availability it is conceivable that strong economies of scale allow to increase societal water supply at little extra cost. But it is unlikely that this cost is zero. However, this is not what the respective statement in the manuscript refers to. The underlying assumption here is that severe hydrological change will increase the cost for maintaining or increasing societal water supply (see Methods). The smaller the number of people affected by severe hydrological change the smaller these costs. This does not mean that achieving water-related SDGs will come at no cost, it just means that avoiding severe hydrological change will not make them more expensive.

Changes in manuscript: Rephrase and clarify.

*Comment: Pg10ln3 "one quarter of the total population" I am a bit confused with this statistic following the 11% figure cited in the prior paragraph.*

Response: The first figure (11 %) is the percentage of people affected at 2 °C. "One quarter of total population" refers to the percentage of people affected at 2.5 °C.

Changes in manuscript: None.

*Comment: Pg10ln14 a version of Fig S6 should be included in the main article.*

Response: Table 1 is in fact a full representation of the panel SSP2/ANY in Fig. S6. We do not think that detailed results for MAD, ND, and Q10 or additional SSPs are required in the main paper. Changes in manuscript: None.

*Comment: Pg11ln16 the meaning/intent of "disproportionally strong" is unclear*

Response: It means that severe hydrological change tends to occur in places affected by water scarcity.

Changes in manuscript: Rephrase and clarify.

*Comment: Pg11ln19 suggest "affect populations [already coping with water scarcity]. Since the specific affected populations have not been identified or discussed, I do not believe a statement about "room for further adaptation" can be supported here.*
Changes in manuscript: Apply proposed change.

*Comment: Pg11ln22 the phrase "more likely than not" should be italicized and applied more methodically throughout where appropriate.*
Changes in manuscript: Check and improve consistent use of 'more likely than not' and related terms throughout the manuscript.

*Comment: Pg11ln22-23 suggest "hydrological change. [Of those affected, 1.9 billion (21.2% of global population) would have. . ."*
Changes in manuscript: Apply proposed change.

*Comment: Pg11ln23 "limited capacity to adapt". Again, I do not believe the study supports statements about barrier to adaptation. It is unclear that population pressure [always] limits capacity to adapt.*
Response: The assumed relationship between population pressure and capacity to adapt is indeed never justified or explained in the paper.
Changes in manuscript: Clarify with references how population pressure relates to capacity to adapt or remove statement.

*Comment: Pg11ln30 separate statistics should be cited for Latin America and Middle East and North Africa regions*
Changes in manuscript: Apply proposed change.

*Comment: Pg12ln1-2 Is this statement sill in reference to LA and MEA or globally,*

*in-general?*
Response: This refers to LAM and MEA.
Changes in manuscript: Rephrase and clarify.

*Comment: Fig1 the flow unit should be defined in the caption. Why is SSP3 so different in terms of total population growth? I'd personally prefer a figure with panels for each region with pop. Growth and % in each water class.*
Response: Yes, flow units should be defined in the caption or the figure itself. Population growth in the different scenarios is determined by the storyline assigned to these scenarios. SSP3 is an extreme scenario in many aspects of its storyline so it is no surprise that it shows the most extreme population increase. However, the reasons for the differences among population scenarios are not relevant for the understanding of this paper.
Changes in manuscript: Add definition of flow units in the caption or the figure.

*Comment: Fig 2. The specific thresholds that define "critical", i.e. 20% decrease in MAD, 50% increase in ND, and 30% increase in Q10 should be noted here.*
Changes in manuscript: Add definition of severe hydrological change.

*Comment: Fig3. Clarify in the caption that this data corresponds with Fig.2. Why is the population affected by the two classes unvarying? It appears to be 3billion for all cases for <1000 p/fu. Why not just limit the x-axis to 70% for readability? Big takeaways-droughts are impacted at all warming levels, MAD above 2C, and Q10 above 3C?*
Response: Water crowding is calculated for climate representing contemporary conditions and is therefore independent of climate change. Extending the x-axis to 100 % has the benefit of preserving the correct proportion of coloured areas (affected population) to total plot area (total population).
Changes in manuscript: None.

*Comment: Table 1. What is the scale factor for the population? The map of the regions delineated should be included in the main paper. Does the 33% of global population correspond to the "any" category in Fig 2d? Does the 21.6% of global population affected at +5C correspond with the "any" category in Fig 3d? Where would the 65% from Fig 3d fall in this table, if there was an additional section? Clarify that all percentages are provided as a percentage of total global population.*

Response: Population is given in million. We do not think that a map of regions is necessarily required in the main paper since the region names are quite common and can be readily understood. The 33.3% correspond to the total number of people affected by water scarcity (>1000 p/fu) in Fig. 1. Yes, 21.6 % corresponds to the affected population at 5 °C warming in the ANY category in Fig 3d. The 65 % in Fig. 3d are the percentage of people under water scarcity affected by severe hydrological change (1940 out of 2988 million). All percentages are given are percentages of population in the spatial unit denoted in the first column (10 regions and 1 world).

Changes in manuscript: Mention that population numbers are in million and that percentages refer to region/world totals.

*Comment: Fig S1 Most differences appear in Middle East, China, South Asia, and North Africa. The discussion and supporting figures could do better to highlight this finding.*

Changes in manuscript: Improve reference to supporting material in main text.

*Comment: Fig S2 why does the SSP2 scenario no match what is shown in Fig.3?*
Response: Fig. S2 matches Fig. 2.
Changes in manuscript: None.

[Figure]

*Comment: Fig S3 "water [crowding]"*
Changes in manuscript: Apply proposed change.

*Comment: Fig S4 "water [crowding]"*
Changes in manuscript: Apply proposed change.

*Comment: Fig5 this belongs in the main article.*
Response: We do not think that a map of regions (Fig. S5) is necessarily required in the main paper since the region names are quite common and can be readily understood.
Changes in manuscript: None.

*Comment: Fig S6 consider a version of this in the main article. Perhaps 10 panels covering each region?*
Response: We plan to give the regional results more room in the revised manuscript. Perhaps with an additional figure similar to the one proposed.
Changes in manuscript: Give more room to regional results, add figure if appropriate.

*Comment: FIgS7 Why not include this plot and parallel plots for ND and Q10 in the main article?*
Response: This Fig. S7 is a reproduction of a figure in another paper (Schewe et al., 2014) based on our own simulation results. Its sole purpose is to demonstrate how our simulation results relate to the results from a more comprehensive model ensemble in that paper. The way we analyse our results in the main paper, in particular the way we deal with uncertainty (by reporting only results that are more likely than not), differs fundamentally and is not compatible with this kind of arrangement. Thus, we think Fig. S7 is adequately placed in the SI.

Changes in manuscript: None.

**References**

Schaphoff, S., von Bloh, W., Rammig, A., Thonicke, K., Biemans, H., Forkel, M., Gerten, D., Heinke, J., Jägermeyr, J., Knauer, J., Langerwisch, F., Lucht, W., Müller, C., Rolinski, S. and Waha, K. (2018) 'LPJmL4 – a dynamic global vegetation model with managed land – Part 1: Model description', Geoscientific Model Development, 11(4), pp. 1343–1375. doi: 10.5194/gmd-11-1343-2018.

Schaphoff, S., Forkel, M., Müller, C., Knauer, J., von Bloh, W., Gerten, D., Jägermeyr, J., Lucht, W., Rammig, A., Thonicke, K. and Waha, K. (2018) 'LPJmL4 – a dynamic global vegetation model with managed land – Part 2: Model evaluation', Geoscientific Model Development, 11(4), pp. 1377–1403. doi: 10.5194/gmd-11-1377-2018.

Schewe, J., Heinke, J., Gerten, D., Haddeland, I., Arnell, N. W., Clark, D. B., Dankers, R., Eisner, S., Fekete, B. M., Colón-González, F. J., Gosling, S. N., Kim, H., Liu, X., Masaki, Y., Portmann, F. T., Satoh, Y., Stacke, T., Tang, Q., Wada, Y., Wisser, D., Albrecht, T., Frieler, K., Piontek, F., Warszawski, L. and Kabat, P. (2014) 'Multimodel assessment of water scarcity under climate change.', Proceedings of the National Academy of Sciences of the United States of America, 111(9), pp. 3245–50. doi: 10.1073/pnas.1222460110.

---

## Author Comment (AC2) · 29 Jun 2018

We wish to thank the anonymous reviewer for her/his comments. Below, we provide detailed responses and propose changes in the manuscript.

*Comment: It is good that a manuscript is concise. However, in the description of the results I find it not really concise, as in the mean time the description of data used and methodology lacks information. A more detailed description of the latter is needed to better understand the results.*

Response: We will improve the presentation of results and the description of methods in the revised manuscript. Reviewer 1 has provided a number of detailed suggestions

how the manuscript can be improved in this regard.
Changes in manuscript: Improve methods and results section, clarify main objective of the paper.

*Comment: Climate change projections: What I find particularly missing in the manuscript is the topic of sea level rise. Under all average global temp rise due to climate change, the sea level will rise. The authors discuss temp rise until 5degrees C, so different sea level rises will occur. Why has this not been accounted for in the manuscript? E.g. coastal flooding will occur without appropriate adaptation. Is this included in the 10-year flooding scenario of the authors? Please discuss. Add a new section under section 4 on issues like this and other uncertainty/limitations of the study.*
Response: While coastal flooding due to sea level rise is an important threat to human settlements, it has no link to terrestrial freshwater resources as such. In contrast, river flooding does not only pose a threat to human settlements but also–and that is the rationale for including it in the paper–to water management infrastructure built into or along rivers. However, we see that the link between increases Q10 and potential threats to water management infrastructure (i.e., change in magnitude or return time of Q100 and Q1000 floods) could be made clearer.
Changes in manuscript: Clarify link between increases Q10 and changes in magnitude and/or return time of Q100 and Q1000.

*Comment: The metric MAD, whether or not in combination with the water crowding indicator: The authors write on page 2 lines 22-23 on the important topic of "... seasonal shortages and changes in variability". They quote that it is important to address this. However, by using a metric like MAD, I do not see at all that seasonality or changes in seasonality due to climate change are addressed. Like in mountain regions, winter can have more water availability due to climate change but summer less. In a mean annual*

*metric this is not accounted for. Also in the water crowding indicator this seasonality is not represented. Please discuss, and again in a new section under section 4 "other uncertainty/limitations of the study". A new publication in STOTEN about water stress partly discusses these issues - https://doi.org/10.1016/j.scitotenv.2017.09.056 . Please discuss relating to this paper.*

Response: The inability of MAD metric to account for changes in seasonal and interannual distribution is addressed in the paper by using two additional metrics (ND and Q10) that capture important aspects of such changes (longer drought periods and larger floods). Despite its shortcomings with respect to seasonality, MAD is still a valuable metric which provides insights in the change of mean water availability. We consider the combination of three metrics covering different aspects of hydrological change as one of the main achievements of this paper. Transgression of certain threshold for all three metrics are jointly referred to as 'severe hydrological change' in the paper and are analysed both separately and jointly in the light of global mean temperature increase. With regard to water crowding, we agree that this indicator has its shortcomings but with the current availability of socioeconomic scenarios for the future it is barely possible to compute more complex metrics. It is important to note that the purpose of the water scarcity metrics discussed in the mentioned paper is to monitor progress towards SDG 6.4. This a completely different application that can build on actual observations and statistics. However, discussing the limitations of the water crowding indicator certainly supports the correct interpretation of the results presented in the present paper and will be added in the revised manuscript.
Changes in manuscript: Add discussion of shortcomings of water crowding as a metric for water scarcity.

*Comment: Page 2 Line 4 "the water supply" delete "the"*
Changes in manuscript: Apply proposed change.
*Comment: Page 2 Lines 32-34: "more significant". Why? I do not see this. Why is this change in water scarcity more significant in already stressed than unstressed regions. Is an increase in water stress not important in any region? Are water users and the environment not affected in both situations? Please justify this statement or alter it. * Page 3 Lines 2-4: same comment*

Response: Indeed, severe hydrological change will always affect people and the environment. But in a situation with ample water availability has enough leeway to compensate for the negative effects on societal water supply. When all water is already appropriated this is much harder. However, 'more significant' may not be the best wording to describe this relationship.

Changes in manuscript: Rephrase and clarify.

*Comment: Population growth: give more information on quantities and assumptions in the scenario's used*

Response: SSP storylines are described in the relevant literature and are not important for the understanding of the results in the present paper. However, we will add a paragraph that mentions names of the scenario and characterizes them briefly.

Changes in manuscript: Add section with short SSP scenario description.

*Comment: Page 7 Lines 1-17: Does the water crowding-indicator account for ground water and environmental flows? Please discuss, again referring to https://doi.org/10.1016/j.scitotenv.2017.09.056*

Response: Renewable groundwater is implicitly accounted for in the discharge figures from LPJmL. Environmental flows are not accounted for. We will clarify these aspects in the manuscript.

Changes in manuscript: Add a note on groundwater and environmental flows.

*Comment: page 8 line 11: increase in MAD, or is it decrease? Discuss in more detail -*

*more precipitation but also ET, so what happens with resulting MAD*
Response: This is a typo, it should read: "[. . .] affected by a severe decrease in MAD, [. . .]". Potential drivers of hydrological change will be discussed in the introduction.
Changes in manuscript: Correct typo, add discussion of drivers of hydrological change.

---

## Author Response (AR1)

**List of relevant changes in the manuscript**

1. Large parts of the manuscript have been rewritten to shift the focus of the paper more towards the analysis of severe hydrological change. The main purpose of that is better communicate our key findings, which have remained unchanged.

2. The rational for distinguishing severe hydrological change by different water scarcity classes (based on WCI) that would occur through population change only (without additional climate change) was slightly modified. We now argue (and support by appropriate references) that adaptation to severe hydrological change is more challenging under prevailing water scarcity because it has to include demand-side management strategies, which are faced with bigger obstacles for implementation.

3. An error in the calculation has been found and corrected. Different flow direction maps were used for the LPJmL simulations and for the calculation of grid cell WCI. Using the same flow direction dataset (STN-30) results in slightly larger estimates of people affected by water scarcity but does not impact the key findings of the paper.

4. Instead of estimating Q10 based on maximum monthly discharge in each year we now use 5-day average annual peak flow, which is more in line with standard hydrological procedures. This results in slightly lower estimates of population *more likely than not* exposed to severe increases in flooding hazard and severe hydrological change but has no impact on the key findings of the paper.

5. We have clarified that Q10 serves as a proxy for changes in large floods, which are more relevant from water resource perspective. We have added a section where we evaluate the accuracy of using Q10 to detect changes in large floods and demonstrate that the threshold of 30 % is conservative in that it produces more false negatives than false positives.

6. A short description of the spatially explicit SSP population scenarios has been added.

7. A discussion of the limitation of using the three selected metrics and their thresholds to detect severe hydrological change has been added. The limitations of the WCI as a water scarcity indicator are discussed, too.

8. Several minor changes have been done to the text and the figures to improve readability and understanding.

9. Fig. S3 has been removed because we do not refer to it in the revised manuscript.

**Point-by-point response to review #1**

*Comment: Although I am not a water manager or decision maker, I do not see that the information provided is particularly actionable. The regions are quite large. Can the authors provide a subset of the most striking examples of sub-regions under population and hydrologic change stress (i.e., map the areas of Table 1 for each degree warming?).*

Response: The main objective of the paper is to inform the climate mitigation discussion with an emphasis on the implications of success and failure of the Paris climate agreement. The focus is therefore on temperature levels at which transgression of critical thresholds may occur as well as on the number of people affected at different temperature levels. This kind of information may not be particularly actionable for water managers, but it is not the paper's intention to provide that. In this light, the table with regionally affected population merely serves a breakdown of global aggregated affected population presented before, to highlight that: (i) affected population is unevenly distributed around the globe; and (ii) the extent to which numbers of affected population increase with temperature also differs greatly across regions. A further breakdown into smaller sub-regions would bear the risk of obscuring these massages by adding too much detailed information. Also, the second aspect would be difficult to see in a set of maps.

Changes in manuscript: We have rewritten large parts of the manuscript to sharpen the key messages in the paper and to state the objective of the paper more clearly.

*Comment: The authors also do not discuss regional uncertainty in the GCMs or SSPs. For example, Fig. S7 shows (white) areas where there is disagreement among models and conversely, agreement among models. I would think at least the model spread (as a surrogate for projection uncertainty) should be discussed in-hand with the water stress projections.*

Response: In this paper, GCM uncertainty is addressed by analysing only impacts that are 'more likely than not'. This is also the case for the regional results in Table 1. Analysing model spread represents an alternative way of dealing with projection uncertainty and has been extensively applied in existing literature. We consider the approach taken in this paper to be more suitable with the intended objective to inform the climate mitigation discussion. The model spread shown in Fig. S7 only serves as a basis for comparison of our projections with existing literature, namely Schewe et al., 2014. The impact of population scenario uncertainty has been taken into account explicitly by analysing results for all five SSP population scenarios. However, since our findings show that most aspects of population exposure to severe hydrological change do not vary much across population scenarios, we have decided to focus on the medium scenario SSP2 in the main paper and show detailed results for all other SSPs in the SI only.

Changes in manuscript: We have tried to make it more clear that alternative population scenarios are only used to demonstrate that our key findings do not depend on the choice of the population scenario.

*Comment: I also misunderstand the statistics for "ANY" hydrologic changes (e.g., Fig 2 and throughout). Take the Indian subcontinent, for example, which is colored in panel (d) but not for (a-c). How can the "ANY" affected area exceed the sum of the area for the 3 metrics of severe change?*

Response: Fig. 2d shows the temperature for each pixel, where severe hydrological change of any of the three types occurs in 10 out of 19 GCMs. For example, this could be a combination of 5 GCMs showing transgression in MAD and 6 other GCM showing transgression in ND, which booth would not show up in the individual maps for these metrics.

Changes in manuscript: We have renamed "ANY" to "severe hydrological change" and have a stronger focus on it. From how we relate severe hydrological change to it's components MAD, ND, and Q10, it should be clear how it should be interpreted.

*Comment: The "majority", "more likely than not" and other terminology was introduced but not consistently or clearly applied throughout the paper.*

Changes in manuscript: We consistently use 'more likely than not' throughout the revised manuscript.

*Comment: Abstract: the fact that MAD, ND, and Q10 are used to quantify variability needs explicit mention.*

Response: Although we regard the consideration of variability related hydrological change as an important contribution, we do not think it needs to be emphasized in the abstract. Instead, we believe the paper will benefit from a stronger alignment along the term 'severe hydrological change', which summarizes severe changes in aspects that are related to variability (ND and Q10) and aspects that are not (MAD).

Changes in manuscript: We rewrote large parts of that paper to emphasize the analysis of 'severe hydrological change'. We introduce the term 'severe hydrological change' in the abstract and provide a detailed motivation and definition of its components in the introduction and the methods. A large part of the results section is concerned with the relationship between severe hydrological change in general and its components MAD, ND, and Q10.

*Comment: Ln9 suggest something closer to the following: "by water scarcity. [Simultaneously, global warming is shifting the seasonality and overall quantity of available water. This study estimates the separate and joint effects of population growth and hydroclimate change on global water resources for a range of warming scenarios, ranging from +1.5- 5C. Hydroclimate change is quantified through three metrics: mean annual discharge, number of drought months, and magnitude of the ten-year return flood event. . . .and evaluates how climate [change] mitigation. . .hydrological change, as well as the severity of the water scarcity. The results show that without climate [change] mitigation. . ."*

Response: We much appreciate these improvements to the abstract. However, the main focus of our study is on severe hydrological change related to climate change and not on water crowding/scarcity. The latter mainly serves the purpose to highlight where adaptation to severe hydrological change may be particularly challenging. We see that the abstract does not make this clear in the best possible way and we will revise the abstract to improve this aspect.

Changes in manuscript: The abstract has been revised abstract to better convey the objectives and elements of the study.

*Comment: Pg2ln1-5 there should be discussion of changing precipitation phase (snow vs rain), earlier springs, longer growing seasons, and glacial melt (Himalayas).*

Response: We agree that the paper would benefit from an elaboration of what we mean by 'changing the hydrological conditions' here.

Changes in manuscript: We have replaced 'changing the hydrological conditions' by some examples of changing hydrological conditions due to climate change.

*Comment: Pg2ln23 it is unclear how the current approach addresses "changes in variability"*

Response: We agree that "changes in variability" is a very broad term and that this sentence requires refinement to express more clearly what is missing in existing analyses and what will be address in this study.

Changes in manuscript: The sentence has been removed.

*Comment: Ph2ln27 the number of drought months does not tell us about the severity of the drought. And multi-year droughts are not identified because every year is treated independently.*

Response: During preparation of the paper we have experimented with various metrics of drought severity but found the outcomes to be very inconsistent and highly dependent on the chosen metric. We therefore decided to use the number of drought months as a metric for changes in droughts, which is simpler and more transparent. ND as calculated here does in fact cover multi-year droughts as the CDM method is applied to the entire continuous time series.

Changes in manuscript: We an explicit statement in the methods that ND is able to capture multi-annual droughts.

*Comment: Pg3ln14 suggest [0.5 x 0.5 degree grid cell]*

Changes in manuscript: Changed as suggested.

*Comment: Pg3ln22-23 where does temperature come from? In cases for which GPCC and CRU precipitation are both available, which one is used, or how are the two estimates merged? These datasets revert to climatology in months for which observations are not available. So, the inter-annual variability of the random sample will be less than actual unless these years are filtered.*

Response: There is a typo in this line, it should read: "[..], time series are based on temperature and cloud cover from CRU TS3.1 [...]." For precipitation, only GPCC is used. The problem that datasets revert to climatology in months where observations are missing, is primarily an issue in the first half of the 20th century in the CRU dataset. For the construction of the reference time series the period 1961 – 2009 was used, and it should therefore be largely unaffected by this

problem. In addition, the months instances where the climatology was used are difficult to identify and their removal would introduce artifacts in the reference time series (only full years and global fields were resampled to retain the temporal and spatial autocorrelation of climate data).

Changes in manuscript: Typo corrected.

*Comment: Pg3ln24 suggest "[random] resampling [with replacement]"*

Changes in manuscript: Changed as suggested.

*Comment: Pg3ln27 what are the inputs and parameters for LPJmL?*

Response: The method section provides detailed information about the climate inputs used to drive LPJmL in this study. A paper including a full model description including all standard input data and parameters has just been published (Schaphoff, von Bloh, et al., 2018), along with a comprehensive model validation paper (Schaphoff, Forkel, et al., 2018). At the time of submission of the present manuscript these papers were still pending final publication. Hence, only the corresponding discussion papers of both these papers could be referenced. These refences will be updated in the revised version of the present manuscript.

Changes in manuscript: References of model documentation and validation updated.

*Comment: Pg5ln7 it should be noted that in practice the seasonality of the MAD shortfall matters, since the reservoirs (and rivers) are managed for alternative purposes including flood management and hydroelectric power generation*

Response: Decrease in MAD (mean annual discharge over 30 years) is a metric for the decline in long-term average water availability. It lacks any information of seasonality or interannual variation, which is the very reason why we have complemented it with metrics capturing changes in droughts and floods.

Changes in manuscript: None.

*Comment: Pg5ln16 is this "river discharge" or "grid cell runoff". There is no mention of a routing scheme. Is the CDM approach most needed because the analysis is on a grid-by-grid basis?*

Response: It is river discharge. The routing scheme is in not mentioned, indeed. The CDM approach is needed to identify droughts in ephemeral rivers.

Changes in manuscript: We have included a sentence that describes how LPJmL simulates river discharge and which drainage direction map has been used.

*Comment: Pg5ln26 why not just use the standardized precipitation index-6months (SPI-6)?*

Response: SPI and related indices are used to detect meteorological drought whereas the focus in this paper is on hydrological drought.

Changes in manuscript: None.

*Comment: Pg6ln14 large floods can be important "drought busters". What is the logic for penalizing Q10 floods in a water scarcity context? Could the authors provide further justification for the 30 % threshold? What is the relative frequency shift? i.e., Q10 plus 30 % is what return flood on-average for each region?*

Response: The rationale for including changes in flood hazards is because large floods (with a return time of 100 years or more) pose a threat to water supply infrastructure and because allocating more storage volume of a reservoir for flood protection (leaving it empty) comes at the cost of reduced storage available for water supply. However, estimating changes (either magnitude or return time) of large floods from a 30-year time series involves fitting an appropriate extreme value distribution (Gumbel or GEV) to the data. Good fits could only be obtained for about half of the grid cell in each scenario and GCM and can therefore not be applied here. Therefore, we use changes in the magnitude of Q10 as a proxy for changes in the magnitude of large floods.

Changes in manuscript: We have tried to emphasize the fact that changes in the magnitude in Q10 are only a proxy for changes in the magnitude of large floods. Further, we added a section in which we use the cases where we obtained reasonable fits of a GEV to evaluate the accuracy of using Q10 to detect changes in large floods. Based on this we motivate the threshold for changes in Q10 by the number of falsely detected (false positives) and overlooked severe increases in the magnitude of large floods (false negatives). We show that 30 % is a conservative choice in that it produces more false negatives than false positives.

*Comment: Pg6ln19 the authors have omitted a section introducing SSP scenarios*

Response: SSP storylines are extensively covered in existing literature precise knowledge about these storylines is not needed for the interpretation of results in this paper. Nevertheless, mentioning at least the scenario names is certainly useful.

Changes in manuscript: We have added a short description of the gridded SSP population scenarios.

*Comment: Pg7lns11 what are the "basins" that were used. Can these be illustrated?*

Response: River basins are defined by the drainage network also used for river routing in the model.

Changes in manuscript: Reference to flow direction map added.

*Comment: Pg8ln4-5 it would be useful for the authors to provide figures that better illustrate the relative/shared contribution of both pathways named here.*

Response: The focus of this paper is on population exposure to severe hydrological change due to global warming. The water crowding analysis merely serves as a means to estimate where theses changes may matter the most. We do not think that knowing how much of future water stressed population is caused by each of the two pathways is relevant in this context. In particular, as the amount of information is quite large (5 SSPs times 5 crowding classes). In addition, it may not be so easy to disentangle the two cases as population will often continue to grow after a pixel has tipped into water stress or scarcity.

Changes in manuscript: Statement removed.

*Comment: Pg8ln8 define "substantial"*

Response: While we agree this is a rather vague term, a more precise wording is not required at this point. Detailed information on population affected by severe hydrological change at different levels of global warming is provided in the remainder of this section.

Changes in manuscript: Removed.

*Comment: Pg8ln9 what is "severe"*

Response: Thresholds for what consist a severe change in MAD, ND, and Q10 are defined and motivated in the method section. However, we see that it is not entirely clear that these thresholds are referred to here.

Changes in manuscript: The paper has been carefully rewritten to assure a consistent use of 'severe change'. Threshold are defined and motivated in the methods and also given in Fig. 1 again.

*Comment: Pg8lns17-19 are these statistics from a table or figure or not shown? Is it 108 million fewer or total impacted? 319 million fewer or total? 15 million fewer or total? Unclear if "these figures" are total affected or differences from 5C numbers is prior paragraph.*

Response: All numbers mentioned in this section are displayed in Fig. 2. All three numbers mentioned by the reviewer are totals not differences.

Changes in manuscript: The paper has been carefully rewritten to avoid such ambiguities.

*Comment: Pg8lns16-30 is there no feedback between the population scenario and water availability? Will population continue to grow as projected despite severe shortages of water?*

Response: Population projections are part of the SSP scenario sets. They do not account for climate change impacts. Neither does the downscaling of national values to grid cell.

Changes in manuscript: We added a clarifying sentence in the description of gridded population scenario in methods.

*Comment: Pg9ln2 "water supply systems" statement is vague and should be qualified by references*

Changes in manuscript: Removed.

*Comment: Pg9ln9 where regionally are these 2.99 billion people concentrated?*

Response: This can be seen in the first column of Table 1 but is not explicitly mentioned in the text since the focus of the paper is on severe hydrological change and not on water crowding.

Changes in manuscript: None.

*Comment: Pg9ln11 where geographically is this change in high crowding projected?*

Response: This seems to be a misunderstanding: water crowding/scarcity is a static overlay in our analysis, calculated from river discharge under contemporary climate conditions and future population patterns (here SSP2). We will make sure to make this clearer in the revised manuscript. The regional distribution of population experiencing water scarcity and severe hydrological change is given in Table 1 and discussed in the main text.

Changes in manuscript: The paper has been rewritten in large parts to better describe the relevance of water crowding and how it is calculated.

*Comment: Pg9ln13 "occur in places", please provide examples.*

Response: This not as easy as it seems. Water crowding often occurs in places of small spatial extent (a few grid cells) but high population density (i.e., cities). We do not think that the scale of analysis (global model, pattern scaled climate scenarios, 0.5 degree resolution) allows to report results at the scale of cities. Regional examples are given in Table 1 and are discussed in the text.

Changes in manuscript: None/removed.

*Comment: Pg9ln18 I misunderstand. How is ANY (1.94billion) larger than the sum of MAS, ND, and Q10?*

Response: This is because in the ANY category, population is in at least 10 out of 19 GCM exposed to a severe hydrological of any kind. Thus, all combinations of severe for MAD, ND, and Q10 are possible here. For example, a transgression of MAD in 5 GCMs and a transgression of ND in 6 other GCM, would mean a pixel and its population is considered in the ANY category but neither in the MAD nor the ND category. While we have tried to provide a definition of ANY in

the paper (p8ln25-28), we acknowledge that correct understanding is crucial and deserves more space.

Changes in manuscript: There is stronger focus on 'severe hydrological change' (the ANY metric) in the revised manuscript and it should be clearer now how it relates to the individual metrics. The difference between the combined metric (ANY) and the sum of the individual metrics (MAD, ND, Q10) is explicitly quantified and explained.

*Comment: Pg9ln21 it would be instructive to see the regional distribution of the projected benefit of the Paris agreement*

Response: This is shown in Table 1 and described in the following section.

Changes in manuscript: None

*Comment: Pg9ln22 intended reference to "remaining number of people affected by severe hydrological change" is unclear.*

Changes in manuscript: We have clarified that this refers to a 2° C warming.

*Comment: Pg9ln26 clarify here if "11 % of total population" is of global affected population (274mil) or LAM or MEA population*

Changes in manuscript: We have clarified that this refers to the population in the respective regions.

*Comment: Pg9ln26 is this "another 29 % [of the affected global population] live in SAS and SSA, although they locally comprise only 2 % of the population"?*

Response: This is correct.

Changes in manuscript: Clarification added.

*Comment: Pg9ln28 "since substantial societal and economic efforts". Please clarify or add a reference.*

Changes in manuscript: Statement removed.

*Comment: Pg9ln31-32 "reduce the costs" this may not always hold. Depending on the region and the solution, the infrastructure investment to serve 5 % may be just as expensive as serving 10 %.*

Response: Under conditions of unconstrained water availability it is conceivable that strong economies of scale allow to increase societal water supply at little extra cost. But it is unlikely that this cost is zero. However, this is not what the respective statement in the manuscript refers to. The underlying assumption here is that severe hydrological change will increase the cost for maintaining or increasing societal water supply (see Methods). The smaller the number of people affected by severe hydrological change the smaller these costs. This does not mean that achieving water-related SDGs will come at no cost, it just means that avoiding severe hydrological change will not make them more expensive.

Changes in manuscript: This statement has been remove in favour of different line of argument emphasizing adaptation challenges rather than cost.

*Comment: Pg10ln3 "one quarter of the total population" I am a bit confused with this statistic following the 11 % figure cited in the prior paragraph.*

Response: The first figure (11 %) is the percentage of people affected at 2 °C. "One quarter of total population" refers to the percentage of people affected at 2.5 °C.

Changes in manuscript: Clarified.

*Comment: Pg10ln14 a version of Fig S6 should be included in the main article.*

Response: Table 1 is in fact a full representation of the panel SSP2/ANY in Fig. S6. We do not think that detailed results for MAD, ND, and Q10 or additional SSPs are required in the main paper.

Changes in manuscript: None.

*Comment: Pg11ln16 the meaning/intent of "disproportionally strong" is unclear*

Response: It means that severe hydrological change tends to occur in places affected by water scarcity.

Changes in manuscript: Removed.

*Comment: Pg11ln19 suggest "affect populations [already coping with water scarcity]. Since the specific affected populations have not been identified or discussed, I do not believe a statement about "room for further adaptation" can be supported here.*

Changes in manuscript: Sentence removed.

*Comment: Pg11ln22 the phrase "more likely than not" should be italicized and applied more methodically throughout where appropriate.*

Changes in manuscript: We use "more likely than not" more consistently and italicized in the revised manuscript.

Comment: *Pg11ln22-23 suggest "hydrological change. [Of those affected, 1.9 billion (21.2 % of global population) would have..."*

Changes in manuscript: Rephrased.

Comment: *Pg11ln23 "limited capacity to adapt". Again, I do not believe the study supports statements about barrier to adaptation. It is unclear that population pressure [always] limits capacity to adapt.*

Response: The assumed relationship between population pressure and capacity to adapt is indeed never justified or explained in the paper.

Changes in manuscript: We have clarified and support with references that adaptation under water scarcity is more difficult because it has to include demand-side management strategies, which are faced with bigger obstacles for implementation.

Comment: *Pg11ln30 separate statistics should be cited for Latin America and Middle East and North Africa regions*

Changes in manuscript: Change as suggested.

Comment: *Pg12ln1-2 Is this statement sill in reference to LA and MEA or globally, in-general?*

Response: This refers to LAM and MEA.

Changes in manuscript: Rephrased and clarified.

Comment: *Fig1 the flow unit should be defined in the caption. Why is SSP3 so different in terms of total population growth? I'd personally prefer a figure with panels for each region with pop. Growth and % in each water class.*

Response: Yes, flow units should be defined in the caption or the figure itself. Population growth in the different scenarios is determined by the storyline assigned to these scenarios. SSP3 is an extreme scenario in many aspects of its storyline so it is no surprise that it shows the most extreme population increase. However, the reasons for the differences among population scenarios are not relevant for the understanding of this paper.

Changes in manuscript: Definition of flow units and WCI classed added in figure.

*Comment: Fig 2. The specific thresholds that define "critical", i.e. 20 % decrease in MAD, 50 % increase in ND, and 30 % increase in Q10 should be noted here.*

Changes in manuscript: Respective thresholds added to figure.

*Comment: Fig3. Clarify in the caption that this data corresponds with Fig.2. Why is the population affected by the two classes unvarying? It appears to be 3billion for all cases for <1000 p/fu. Why not just limit the x-axis to 70 % for readability? Big takeaways- droughts are impacted at all warming levels, MAD above 2C, and Q10 above 3C?*

Response: Water crowding is calculated for climate representing contemporary conditions and is therefore independent of climate change. Extending the x-axis to 100 % has the benefit of preserving the correct proportion of coloured areas (affected population) to total plot area (total population).

Changes in manuscript: Reference to Fig.2 added in caption of Fig. 2.

*Comment: Table 1. What is the scale factor for the population? The map of the regions delineated should be included in the main paper. Does the 33 % of global population correspond to the "any" category in Fig 2d? Does the 21.6 % of global population affected at +5C correspond with the "any" category in Fig 3d? Where would the 65 % from Fig 3d fall in this table, if there was an additional section? Clarify that all percentages are provided as a percentage of total global population.*

Response: Population is given in million. We do not think that a map of regions is necessarily required in the main paper since the region names are quite common and can be readily understood. The 33.3 % correspond to the total number of people affected by water scarcity (>1000 p/fu) in Fig. 1. Yes, 21.6 % corresponds to the affected population at 5 °C warming in the ANY category in Fig 3d. The 65 % in Fig. 3d are the percentage of people under water scarcity affected by severe hydrological change (1940 out of 2988 million). All percentages are given are percentages of population in the spatial unit denoted in the first column (10 regions and 1 world).

Changes in manuscript: Unit of population numbers reference point of percentages clarified in caption.

*Comment: Fig S1 Most differences appear in Middle East, China, South Asia, and North Africa. The discussion and supporting figures could do better to highlight this finding.*

Response: Because the focus of the revised manuscript has shifted even more toward the analysis of sever hydrological change, we have decided against discussing differences between population scenarios here.

Changes in manuscript: None.

Comment: *Fig S2 why does the SSP2 scenario no match what is shown in Fig.3?*

Response: Fig. S2 matches Fig. 2.

Changes in manuscript: None.

*Comment: Fig S3 "water [crowding]"*

Changes in manuscript: Fig. S3 has been removed.

*Comment: Fig S4 "water [crowding]"*

Changes in manuscript: "Water crowding" has been removed.

*Comment: Fig5 this belongs in the main article.*

Response: We do not think that a map of regions (Fig. S5) is necessarily required in the main paper since the region names are quite common and can be readily understood.

Changes in manuscript: None.

*Comment: Fig S6 consider a version of this in the main article. Perhaps 10 panels covering each region?*

Response: As stated above, the purpose of the regional breakdown is to show that substantial risks remain in some regions even for low levels of warming. Table 1 is sufficient for that.

Changes in manuscript: None.

*Comment: FIgS7 Why not include this plot and parallel plots for ND and Q10 in the main article?*

Response: This Fig. S7 is a reproduction of a figure in another paper (Schewe et al., 2014) based on our own simulation results. Its sole purpose is to demonstrate how our simulation results relate to the results from a more comprehensive model ensemble in that paper. The way we analyse our results in the main paper, in particular the way we deal with uncertainty (by reporting only results that are more likely than not), differs fundamentally and is not compatible with this kind of arrangement. Thus, we think Fig. S7 is adequately placed in the SI.

Changes in manuscript: None.

**Point-by-point response to review #1**

*Comment: It is good that a manuscript is concise. However, in the description of the results I find it not really concise, as in the mean time the description of data used and methodology lacks information. A more detailed description of the latter is needed to better understand the results.*

Response: We will improve the presentation of results and the description of methods in the revised manuscript. Reviewer #1 has provided a number of detailed suggestions how the manuscript can be improved in this regard.

Changes in manuscript: We have rewritten large parts of the paper to improve the description of methods and presentation of results.

*Comment: Climate change projections: What I find particularly missing in the manuscript is the topic of sea level rise. Under all average global temp rise due to climate change, the sea level will rise. The authors discuss temp rise until 5degrees C, so different sea level rises will occur. Why has this not been accounted for in the manuscript? E.g. coastal flooding will occur without appropriate adaptation. Is this included in the 10-year flooding scenario of the authors? Please discuss. Add a new section under section 4 on issues like this and other uncertainty/limitations of the study.*

Response: While coastal flooding due to sea level rise is an important threat to human settlements, it has no link to terrestrial freshwater resources as such. In contrast, river flooding does not only pose a threat to human settlements but also–and that is the rationale for including it in the paper–to water management infrastructure built into or along rivers. However, we see that the link between increases Q10 and potential threats to water management infrastructure (i.e., change in magnitude or return time of Q100 and Q1000 floods) could be made clearer.

Changes in manuscript: We have clarified that changes in Q10 are used as a proxy for changes in the magnitude of large floods (with a return time of 100 year or more). We have included a paragraph where we evaluate how changes in the magnitude of Q10 are linked to changes in magnitude of Q100 and Q1000.

*Comment: The metric MAD, whether or not in combination with the water crowding indicator: The authors write on page 2 lines 22-23 on the important topic of "... seasonal shortages and changes in variability". They quote that it is important to address this. However, by using a metric like MAD, I do not see at all that seasonality or changes in seasonality due to climate change are addressed. Like in mountain regions, winter can have more water availability due to climate change but summer less. In a mean annual metric this is not accounted for. Also in the water crowding indicator this seasonality is not represented. Please discuss, and again in a new section under section 4 "other uncertainty/limitations of the study". A new publication in STOTEN about water stress partly discusses these issues - https://doi.org/10.1016/j.scitotenv.2017.09.056 . Please discuss relating to this paper.*

Response: The inability of MAD metric to account for changes in seasonal and interannual distribution is addressed in the paper by using two additional metrics (ND and Q10) that capture important aspects of such changes (longer drought periods and larger floods). Despite its shortcomings with respect to seasonality, MAD is still a valuable metric which provides insights in the change of mean water availability. We consider the combination of three metrics

covering different aspects of hydrological change as one of the main achievements of this paper. Transgression of certain threshold for all three metrics are jointly referred to as 'severe hydrological change' in the paper and are analysed both separately and jointly in the light of global mean temperature increase. With regard to water crowding, we agree that this indicator has its shortcomings but with the current availability of socioeconomic scenarios for the future it is barely possible to compute more complex metrics. It is important to note that the purpose of the water scarcity metrics discussed in the mentioned paper is to monitor progress towards SDG 6.4. This a completely different application that can build on actual observations and statistics. However, discussing the limitations of the water crowding indicator certainly supports the correct interpretation of the results presented in the present paper and will be added in the revised manuscript.

Changes in manuscript: The focus in the revised manuscript has shifted even more towards the analysis of severe hydrological change. It should be even clearer now that the sole purpose of the assessment of water scarcity based on the WCI is to provide a rough classification of adaptation challenges. We briefly discuss the limitations of the WCI as a water scarcity indicator.

*Comment: Page 2 Line 4 "the water supply" delete "the"*

Changes in manuscript: Changed as suggested.

*Comment: Page 2 Lines 32-34: "more significant". Why? I do not see this. Why is this change in water scarcity more significant in already stressed than unstressed regions. Is an increase in water stress not important in any region? Are water users and the environment not affected in both situations? Please justify this statement or alter it. * Page 3 Lines 2-4: same comment*

Response: Indeed, severe hydrological change will always affect people and the environment. But in a situation with ample water availability has enough leeway to compensate for the negative effects on societal water supply. When all water is already appropriated, demand management options need to be considered, which are more difficult to implement. However, 'more significant' may not be the best wording to describe this relationship.

Changes in manuscript: The whole paragraph has been rewritten to clarify the assumed relationship between water scarcity and adaptation challenges.

*Comment: Population growth: give more information on quantities and assumptions in the scenario's used*

Response: SSP storylines are described in the relevant literature and are not important for the understanding of the results in the present paper. However, we will add a paragraph that mentions names of the scenario and characterizes them briefly.

Changes in manuscript: A short section describing the spatially explicit SSP population scenarios has been included.

*Comment: Page 7 Lines 1-17: Does the water crowding-indicator account for ground water and environmental flows? Please discuss, again referring to https://doi.org/10.1016/j.scitotenv.2017.09.056*

Response: As described in the paper, groundwater recharge is included in the available freshwater estimates from LPJmL. Because the sole purpose of the WCI is to provide a rough classification of adaptation challenges we only briefly discuss the caveats of the WCI as a water scarcity indicator.

Changes in manuscript: None.

*Comment: page 8 line 11: increase in MAD, or is it decrease? Discuss in more detail - more precipitation but also ET, so what happens with resulting MAD*

Response: This is a typo, it should read: "[...] affected by a severe decrease in MAD, [...]". Potential drivers of hydrological change will be discussed in the introduction.

Changes in manuscript: Statement removed.

[revised manuscript text omitted]

Unlike most global assessments of climate change impacts on water resources, which have employed a measure of water stress like the water crowding index (WCI; Falkenmark 1989) or the withdrawal-to-availability ratio (WTA; Raskin et al. 1996) we here analyse hydrological changes relevant from a water resource perspective directly. This allows us to focus on climate-induced hydrological change alone (unobscured by the effects of population change) and to include aspects of hydrological change from water resource perspective other than mean annual discharge, on which both WCI and WTA are based. In order to gain a detailed and comprehensive understanding of changes in the water sector, this study analyses climate impacts with respect to decrease in mean water availability, growing prevalence of hydrological droughts, and increase of flooding hazards. To estimate these hydrological changes, three key metrics are used to assess flow regime changes: (i) mean annual discharge (MAD); (ii) the average number of drought months per year (ND); and (iii) the 10-year flood peak (Q10). Severe hydrological

change is defined as crossing a critical threshold (defined below) for at least one of these key metrics. By combining these changes with spatially explicit population projections consistent with Shared Socioeconomic Pathways (SSPs) (Jones and O'Neill 2016), the number of people exposed to severe hydrologic changes is estimated for each level of $\Delta T_{glob}$.

However, looking at the total number of people affected by severe hydrological change provides only limited insights about the consequences of severe hydrological change and the challenges for adaptation. These are greatly determined by the underlying population-driven water scarcity level – that is, when options for supply-side management are exhausted or become too costly under water scarcity conditions, the focus of water management has to shift towards demand management (Falkenmark 1989; Ohlsson and Turton 1998). Thus, adaptation to severe hydrological change under already water scarce conditions will also have to involve demand-side management strategies to prevent negative social and economic consequences. Because demand-side options are complex and their implementation faced with behavioural, economic, political, and institutional obstacles (Russell and Fielding 2010; Kampragou, Lekkas, and Assimacopoulos 2011), adaptation to severe hydrological change is more challenging challenges under already water scarce conditions. To account for this aspect, we apply the WCI to estimate the future population pressure on water resources under the assumption of no climate change and jointly analyse climate-induced severe hydrological change and population-driven water scarcity.

**2 Methods**

**2.1 Population scenarios**

For the estimation of future population affected by severe hydrological change and to calculate WCI, we use spatially explicit population projections from Jones and O'Neill (2016). These are based on the SSP national population projections (KC and Lutz 2017) and have been downscaled making additional assumptions on urbanisation consistent with the respective SSP storyline. The five SSP storylines are designed to cover a broad range of future socioeconomic development pathways with plausible future changes in demographics, human development, economy, institutions, technology, and environment (O'Neill et al. 2017). However, they do not account for the impact of climate change on development pathways. For this study we only use the population projections of the SSPs. The analysis focuses on the middle-of-the-road scenario SSP2 (with a total population of 9.0 billion in 2100), but we use the other scenarios (with a total population between 6.9 and 12.6 billion in 2100) to test the sensitivity of our findings to different population scenarios.

**2.2 Climate scenarios**

In order to systematically assess climate change impacts on freshwater resources, we use the PanClim climate scenarios described in Heinke et al. (2013). The dataset consists of 8 different scenarios of $\Delta T_{glob}$ obtained with the MAGICC6 model (Meinshausen, Raper, and Wigley 2011) based on greenhouse gas emissions that result in a range of warming levels above pre-industrial (~1850) conditions from 1.5 °C to 5.0 °C in steps of 0.5 °C in 2100 (2086–2115 average). For each $\Delta T_{glob}$

pathway, the local response in climate variables is emulated for 19 different General Circulation Models (GCMs) from the Coupled Model Intercomparison Project phase 3 (CMIP3) ensemble using a pattern-scaling approach. In so doing, normalized climate anomalies (changes per 1.0 °C of $\Delta T_{glob}$ increase) of temperature, precipitation and cloud cover for each month of the year in each 0.5 x 0.5 arc-degree grid cell are obtained by linear regression between time series of climate variables and the corresponding time series of $\Delta T_{glob}$. The unexplained variance of these linear models is in the same order (temperature and cloud cover) or only slightly larger (precipitation) than inter-annual variability in the pre-industrial control run without anthropogenic forcing, indicating that most of the climate change information is captured by the obtained patterns. The normalized climate anomalies are used to calculate local climate anomalies for any given $\Delta T_{glob}$ relative to the year 2009 (when $\Delta T_{glob}$ was 0.9 °C above pre-industrial level). These local climate anomalies were then applied to monthly reference time series of local climate that represent average conditions and variability in 2009.

A total of 152 climate scenarios (8 $\Delta T_{glob}$ pathways x 19 GCM patterns) for the period 1901–2115 are obtained. Up to the year 2009, time series are based on mean air temperature and cloud cover from CRU TS3.1 (Harris et al. 2014) and precipitation from the GPCC full reanalysis dataset version 5 (Schneider et al. 2014). The reference time series for the period 2010–2115 to which climate anomalies are applied is created from the historic datasets by random resampling with replacement. Further details on the climate scenarios have been described by Heinke et al. (2013).

**2.3 Impact model**

For assessing the impacts of climate change on the hydrological cycle, we employ the LPJmL Dynamic Global Vegetation Model version 4 (LPJmL4) that simulates the growth of natural vegetation and managed land in coupling with the global carbon and hydrological cycle (Schaphoff, von Bloh, et al. 2018). The model has been extensively evaluated showing good performance in representing the global hydrological cycle (Rost et al. 2008; Schaphoff, Forkel, et al. 2018). LPJmL has been widely applied in water resources assessments (D. Gerten et al. 2011; Rockström et al. 2014; Steffen et al. 2015; Jägermeyr et al. 2016).

For the simulations conducted here, the model is first run without land use for a spinup period of ~5000 years using pre-industrial atmospheric $CO_2$ concentrations and climate data from 1901–1930. This is followed by a second spinup of 390 years up to 2009, during which atmospheric $CO_2$ concentrations and climate vary according to historical observations, and constant land use of the year 2000 is prescribed (Fader et al. 2010). All 152 scenario simulations are initialized from this state, assuming constant land use over the whole simulation period and atmospheric $CO_2$ concentrations consistent with the respective $\Delta T_{glob}$ scenario (Heinke et al. 2013). All simulations are performed without direct anthropogenic intervention on freshwater resources (water withdrawals and dam operation) as their effect are assumed to be captured by the WCI.

In addition to the 152 $\Delta T_{glob}$ scenarios one additional simulation for the period 2010–2115 is carried out using the reference climate data without any anomalies applied and with constant atmospheric $CO_2$ concentrations of the year 2009. This simulation represents a no climate change setting, for which transient time series with inter-annual variability but without a

For the simulations conducted here, the model is first run without land use for a spinup period of ~5000 years using pre-industrial atmospheric $CO_2$ concentrations and climate data from 1901–1930.

general trend are produced. This scenario is used as the reference simulation for the comparison with the other climate scenarios. Because the sequence of dry and wet years is identical in all scenarios and the reference case, any differences between the scenarios and the no-climate-change reference simulation can be attributed to global warming. Within this paper we analyse the 30-year time period from 2086 to 2115, in which the average temperature increase equals $\Delta T_{glob}$.

**2.4 Hydrological change metrics**

The focus of this study is on hydrological changes due to climate change that are relevant from a water resource perspective. "Water" resources" refers to 'blue' water—the water that can be withdrawn from rivers, lakes and aquifers, and which can be directly managed by humans—as opposed to 'green' water, i.e. the soil moisture in the root zone from local precipitation that can only be used by locally growing plants (Rockström et al. 2014).

We here use river discharge as an approximation of the blue water resource. River discharge is simulated in LPJmL by means of linear storage cascade (Schaphoff, von Bloh, et al. 2018) along a river network defined by the STN-30p flow direction map (Vörösmarty et al. 2000). The simulated discharge of a grid cell includes all the water that enters the cell from upstream areas and all surface and subsurface runoff generated within the cell. Although water is often withdrawn from lakes and aquifers, no more than the possible recharge to these storages can be withdrawn over a prolonged period. Therefore, river discharge as computed with LPJmL represents a good approximation of the total renewable blue water resource (excluding non-renewable fossil groundwater from aquifers with very long recharge times).

Three metrics relevant from a water resource perspective, i.e. mean annual discharge (MAD), the number of drought months per year (ND), and the 10-year flood peak (Q10), are calculated for each grid cell for the 8 levels of $\Delta T_{glob}$ and 19 GCM patterns. Severe hydrological change is defined as crossing a critical threshold for at least one of these three metrics: a greater than 20% decreases in MAD, an increase of 50% in ND, and an increase in Q10 by 30% (further described below). Based on these results we determine in each grid cell the lowest level of $\Delta T_{glob}$ at which the thresholds for each of the metrics are transgressed in more than 50 % of GCM runs (at least 10 out of 19). This transgression in more than 50 % of GCMs corresponds to the *more likely than not* likelihood category used in IPCC AR5 (Mastrandrea et al. 2011).

**2.4.1 Mean water availability**

Changes in mean annual discharge (MAD) are used as a measure for changes in mean water availability, assuming that a substantial decline in MAD will make it difficult to satisfy existing and future societal water demands with existing water supply infrastructure. We define a decrease in MAD by 20 % or more as a severe hydrological change that requires some form of management intervention (either on the supply or the demand side). The same threshold was also used by Schewe et al. (2014) to define severe decrease in annual discharge.
* * *
**2.4.2 Hydrological drought**

The occurrence of prolonged periods of below-average discharge, mostly initiated by interannual climate variability, is referred to as hydrological drought. To provide stable water supply to society, water supply systems are adjusted to seasonal variability and drought regimes. A substantial increase in drought periods thus impairs the capability of existing water management infrastructure.

We apply a drought identification method proposed by Van Huijgevoort et al. (2012) to determine which months of a monthly time series of river discharge are in drought condition. The method is based on a combination of the threshold level method (TLM) and the consecutive dry month method (CDM). The TLM method classifies a month as drought-stricken if discharge falls below a given threshold (here, the month-specific discharge that is exceeded 80 % of the time). However, in ephemeral rivers a method that accounts for the duration of dry periods is more appropriate since the TLM would classify all months with zero flow as drought. We adopt this combination of TLM and CDM from Van Huijgevoort et al. (2012) but make some modifications to obtain a more robust and plausible algorithm. First, a month-specific discharge threshold is applied to identify drought months according to the TLM method. Then, if the TLM threshold is zero and the number of drought months in a given calendar month (e.g., January) exceeds 20 %, the CDM is used to determine which of the months with zero discharge can be classified as drought months. To this end, the number of preceding consecutive TLM droughts is determined for each month with zero discharge in the given calendar month. Finally, a threshold is selected that retains only the months with the longest preceding dry period so that the total number of drought months in that calendar month is 20 %. The TLM and CDM thresholds are determined from the reference simulation representing present day climate conditions. These thresholds are then used to estimate the number of drought months for all climate scenarios. Note that the thresholds are derived from and applied to the continuous 30-year time series, which allows for the detection of multi-year droughts.

We define an increase in the average number of drought months per year (ND) by 50 % (i.e., from 20 % to 30 %) as a severe hydrological change that will require an upgrade of existing water management systems to maintain a reliable water supply.

**2.4.3 Flood hazard**

All water supply infrastructure should be designed to withstand typical flooding events. A flood with a return time of the 50-100 years (Q100) is typically used as a reference case (Coles 2001). However, spillways of critical infrastructure such as dams and reservoirs are designed for even more severe flood events, with a return time of 1000 years or more (Dyck and Peschke 1995). An increase in the magnitude of design floods poses a serious threat to water management systems with potentially disastrous consequences.

The magnitude of extreme events with long return periods is usually derived from much shorter observed time series of annual maximum floods by fitting a suitable extreme value distribution (e.g. a Gumbel or Generalized Extreme Value distribution; Coles 2001). The obtained extreme value distribution is then used to extrapolate the magnitude of flood events with long return periods. This procedure can also be used to detect changes in the magnitude or the return time of such events from two fitted

Deleted: A flood with a return time of the 50-100 years (Q100) is typically used as a reference case (Coles 2001) but spillways of critical infrastructure such as dams and reservoirs are designed for flood events with a return time of 1000 years or more (Dyck and Peschke 1995). An increase in the magnitude of these

Deleted: The magnitude of extreme events with long return periods can be derived from much shorter observed time series of annual maximum floods by fitting a suitable extreme value distribution (e.g. a Gumbel or Generalized Extreme Value distribution; Coles 2001). The obtained extreme value distribution is then used to extrapolate the magnitude of flood events with long return periods. Because this procedure is computationally expensive and introduces additional uncertainties, we here analyse changes in the annual maximum flood with a return period of 10 years (Q10). The magnitude of this event is directly derived from the simulated time series of monthly discharge by determining the maximum annual flood that is exceeded in 3 out of 30 years (technically a return time of 10.33 years).
We define an increase in Q10 by 30 % as severe change

extreme value distributions (Dankers et al. 2013). However, fitting a Generalized Extreme Value (GEV) distribution to 5-day average peak flow estimates form LPJmL using *L* moments (Hosking and Wallis 1995) gave good fits (p-value of Kolmogorov-Smirnov test > 0.9) in only about half of all cases. In order to estimate the change in flood hazard for all grid cells, we analyse changes in the magnitude of flood with a 10-year return time (Q10), which are directly derived by determining the 5-day average peak flow that is exceeded in 3 out of 30 years (technically a return time of 10.33 years).

We use the cases where a good fit of the GEV to data was achieved to assess how well the estimated changes in directly derived Q10 can be used as a proxy for changes in events with a higher return time (Q100 or Q1000) derived from GEVs. Because the overall goal is to detect a severe increase in Q100 or Q1000, we estimate how many false positives and false negatives occur when a threshold of 20 % or 30 % increase in Q10 is used. False positives are defined as increases in Q10 by more than 20 % or 30 %, respectively, which does not coincide with an increase in Q100 or Q1000 by at least 10 %; false negatives are defined as an increase in Q100 or Q1000 by more than 50 %, which do not coincide with an increase in Q10 by at least 20 % or 30 %, respectively. For Q100, we find that a threshold of 20 % for Q10 produces 6.3 and 4.7 % of false positives and negatives, respectively; a threshold of 30 % produces 2.6 and 11.0 % of false positives and negatives, respectively. For Q1000 the figures are much higher with 15.9 % (10.7 %) of false positives and 33.8 % (47.0 %) of false negatives for a threshold of 20 % (30 %) for Q10. This demonstrates that Q10 can be used as proxy to detect severe changes in Q100 with reasonable accuracy but not to detect severe changes in Q1000.

We give the avoidance of false positives a higher priority to obtain conservative estimates of flood hazard increase. Therefore, we choose an increase in Q10 by 30 % as a threshold to detect a severe increase in flooding hazard that needs to be addressed by investment in enhancing flood resistance of water supply infrastructure or by changing reservoir operation schemes to increase the safety buffer for flood protection (at the cost of storage capacity for water supply). However, it needs to kept in mind that this indicator only detects about half of the increases in Q1000 by more 50 %, which can be particularly harmful to water management infrastructure.

**2.5 Grid-based water crowding indicator**

In order to determine where transgressions of severe hydrological change thresholds in the three metrics matter most, we estimate which part of global population is experiencing water stress in the absence of additional climate change. We use the WCI originally proposed by Falkenmark (1989) to assess different levels of population pressure on water resources. Originally, the water crowding index was applied at country scale, which may hide important within-country variations (Arnell 2004). With improved spatial distribution of population data and a desire to use natural hydrological units, instead of administrative boundaries, it has become more common to calculate WCI at basin scale (Falkenmark and Lannerstad 2004; Dieter Gerten et al. 2013; Arnell and Lloyd-Hughes 2014; Gosling and Arnell 2016). In this paper, we develop a new calculation procedure to obtain a measure of water crowding that can be calculated and interpreted at grid-cell scale. This can then be combined with the simulated hydrological changes at grid-cell scale to estimate hydrological change for different levels of water crowding.

The simplest way for calculating the grid cell water crowding is by relating total discharge (equivalent to the sum of all upstream and local runoff) to the sum of upstream and local population. Although probably appropriate in many cases, this can lead to an overestimation of crowding (pressure on available water) if a substantial proportion of runoff is generated in parts of the basin with low population. In order to calculate the effective population pressure on the total available water within each grid cell, we therefore treat local (within grid cell) runoff and the inflow from each upstream cell $i$ separately. While local runoff $w_0$ is assumed to be fully available to the local population $p_0$, the inflow from each upstream cell $w_i$ is equally shared between local population $p_0$ and effective upstream population $p'_i$ (eq. 3) corresponding to that inflow:¶

To calculate the effective population pressure on the total available water within each grid cell, we treat local (within grid cell) runoff and the inflow from each upstream cell $i$ separately. The upstream cells of any given grid cell can be derived from the STN-30p flow direction map (Vörösmarty et al. 2000), which is also used to simulate discharge in LPJmL. While local runoff $w_0$ is assumed to be fully available to the local population $p_0$, the inflow from each upstream cell $w_i$ is equally shared between local population $p_0$ and effective upstream population $p'_i$ corresponding to that inflow (eq. 1).

$$w' = w_0 + \sum_{i=1}^{N} w_i \cdot \frac{p_0}{p'_i + p_0} \qquad (1)$$

The obtained effective water quantity $w'$ is the effective available water in that grid cell. Relating local population $p_0$ to $w'$ yields the effective water crowding index $WCI'$ (eq. 2) for the respective cell.

$$WCI' = \frac{p_0}{w'} \qquad (2)$$

Multiplying $WCI'$ with the total water $w$ (sum of local runoff and all inflows) gives the effective population $p'$ (eq. 3) that is required for the calculation of $WCI'$ in the downstream cell.

$$p' = WCI' \cdot w = p_0 \frac{w}{w'} \qquad (3)$$

Because $p'$ of all upstream cells must be known to determine $WCI'$, the calculation for a whole basin starts at the fringes (in cells with no inflow, i.e. where $w_i = p'_i = 0$) and continues consecutively to the basin outlet.

Five different WCI levels can be distinguished, each characterized by a different degree of water scarcity (Falkenmark 1989). WCI below 100 people per flow unit (p/fu; 1 fu = 1e6 m³ per year) are considered uncritical, *quality and dry-season problems* occur between 100 and 600 p/fu, and *water stress* occurs between 600 and 1000 p/fu. Beyond 1000 p/fu a population experiences *absolute water scarcity*, and the level of 2000 p/fu is interpreted as the *water barrier* beyond which all available water resources are utilized. With increasing degrees of water scarcity it becomes progressively harder to fulfil societal water demand by supply-side management and coping with *absolute water scarcity* has to involve demand-side management options (Falkenmark 1989). It is reasonable to assume that adaptation to severe hydrological change under *absolute water scarcity* will not be possible by adjusting water supply infrastructure alone but will also require demand management strategies. Because of the big behavioural, economic, political, and institutional challenges associated with demand management (Russell and Fielding 2010; Kampragou, Lekkas, and Assimacopoulos 2011), we assess exposure to severe hydrological change within the population group experiencing *absolute water scarcity* and within the population group that does not.

**3 Results**

**3.1 Change in water crowding driven by population change**

Between 1950 and 2010 the number of people that live with *absolute water scarcity* (WCI > 1000 p/fu) has increased more than six fold from 295 million (11.7 % of global population) to 1.83 billion (26.8 % of global population) due to population
* * *
growth alone. In the same time period, the number of people beyond the *water barrier* (WCI > 500 p/fu) within that group has increased more than eight fold from 118 million (4.7 % of global population) to 988 million people (14.5 % of global population), so that its share within the group of people living under *absolute water scarcity* has increased from 40.2 % to 53.9 % (Fig. 1c and 1d).

5 This trend is projected to continue in the future under all five SSP population scenarios (Fig. 1c and 1d). The total number of people living under *absolute water scarcity* in 2100 due to population change alone (without any additional climate change) is projected to be higher than today (2010) in all scenarios reaching 2.16 - 5.65 billion (31.5 - 44.9 % of global population), with higher global population being associated with higher absolute and relative numbers of affected people. The number of people who live beyond the *water barrier* is projected to increase to 1.26 – 3.77 billion (18.4 – 29.9 % of global population,
10 58.4 – 66.7 % of population under *absolute water scarcity*).

**3.2 Severe changes in hydrologic conditions under different levels of $\Delta T_{glob}$**

Under the majority of climate change patterns within the range of $\Delta T_{glob}$ considered in this study severe decreases in mean water availability, severe increases in droughts, and severe increases in flood hazard occur in many abundantly populated regions. We estimate that 4.93 billion people (54.9 % of global population) would *more likely than not* be exposed to severe
15 hydrological in the SSP2 population scenario if $\Delta T_{glob}$ reaches 5 °C by 2100 (Fig. 2a; other SSP scenarios see supplementary Fig. S2). Out of these, 1.09 billion, 1.26 billion, and 1.31 billion would *more likely than not* be exposed to a sever decrease in mean water availability, a severe increase in droughts, and a severe increase in flood hazard, respectively (Fig. 2b-d). Note that severe decreases in mean water availability and severe increases in droughts often coincide, which leads to relatively large number of people (889 million) being *more likely than not* exposed to both of these aspects of severe hydrological change. For
20 2.15 billion people, a transgression of the critical threshold for a mix of the three different aspects of severe hydrological change is projected in more than half of the GCMs. The pace at which these levels are reached with increasing $\Delta T_{glob}$ is not linear and differs for the three aspects of severe hydrological change. The additional number of people that become exposed to a severe decrease in mean water availability at each step of $\Delta T_{glob}$ first increases and then declines again, with the by far largest increment occurring between 2 °C and 2.5 °C. A similar pattern is found for exposure to severe increase in droughts with the
25 difference that the largest increase occurs between 1.5 °C and 2 °C. The increment of people becoming exposed to severe increase in flood hazard is very small until 2 °C warming and then steadily increases with $\Delta T_{glob}$. This overall pattern of varying increase in exposure to sever hydrological change with increasing $\Delta T_{glob}$ is very similar across all five SSP population scenarios considered here (Fig. S2).

If global warming was limited to 2 °C by a successful implementation of the Paris Agreement, the number of people *more*
30 *likely than not* exposed to severe hydrological change under SSP2 could be limited to only 615 people (6.9 % of global population; Fig. 2a), protecting almost nine out of ten people (87.5 %) from exposure to sever hydrological change compared to a warming by 5 °C. Because exposure to increased flooding hazard remains very low until 2 °C warming, the majority of

the remaining population would be exposed to severe decreases in mean water availability and sever increases in droughts (Fig. 2b-d). If warming could be limited to 1.5 °C the number of people *more likely than not* exposed to sever hydrological change could be reduced even more to 195 million people (Fig. 2a), a further reduction by more than two thirds (68.4 %) compared to 2 °C warming. However, even a partial failure of the Paris Agreement with an exceedance of the two-degree

5 target by only 0.5 °C would lead to an increase of the number of people exposed to severe hydrological change to 1.14 billion (Fig. 2a)–almost a doubling (84.6 % increase) compared to a warming by 2 °C. The main contribution to this strong increase comes from increased exposure to severe decreases in mean water availability and sever increases in droughts, with exposure to severe increases in flood hazard still playing a minor role at these temperature levels (Fig. 2b-d). Although the total number differ across different population scenarios, the percentage of global population than can be protected from exposure to sever

10 hydrological change by ambitious climate mitigation efforts is very similar across all population scenarios (Fig. S2).

**3.3 Severe hydrological changes and water scarcity**

To get an indication of the adaptation challenges associate with the exposure to severe hydrological change, we use the assessment of future water scarcity due to population change to distinguish two principal adaptation domains. Coping with water scarcity conditions (WCI > 1000 p/fu) even without further aggravation by climate change requires a combination of

15 supply-side and demand-side management measures (Falkenmark 1989; Ohlsson and Turton 1998). Therefore, water demand management interventions will also have to play a role in the adaptation to severe hydrological change under already water scarce conditions. In contrast, adaptation to severe hydrological change under comparatively abundant water availability conditions (WCI ≤1000 p/fu) may be achieved by adjusting water supply infrastructure alone. Although water demand management is generally desirable and may have economic co-benefits (Brooks 2006), it faces many political, legal, and

20 behavioural obstacles for its implementation and may not be practical in all contexts (Russell and Fielding 2010; Kampragou, Lekkas, and Assimacopoulos 2011).

Under the assumption of no climate change, as much as 3.30 billion people (36.8 % of global population) are estimated to live under absolute water scarcity by 2100 in the SSP2 scenario. For all aspects of severe hydrological change and across the whole range of $\Delta T_{glob}$, the proportion of people *more likely than not* exposed to severe hydrological change is much larger in this

25 category than in the rest of the population (Fig. 3). This asymmetric distribution of impacts is most pronounced for severe decreases in mean water availability, severe increases in flood hazard, and for severe hydrological change in general (Figs. 3a, 3c, and 3d). Thus, severe hydrological change is more likely to occur where adaptation may not be possible by adjusting water supply infrastructure alone. This finding is largely independent of the population scenario (Fig. S3).

Because of the challenges associate with the implementation of demand-side management interventions, the population already

30 experiencing water scarcity in the absence of climate change is of primary concern when analysing exposure to severe hydrological change. We estimate that 2.14 billion people (23.9 % of global population) in the SSP2 population scenario would be affected by water scarcity due to population change and *more likely than not* exposed to climate-related severe hydrological

**3.3 Severe changes and water crowding combined**¶
The challenge to adapt water supply systems to severe hydrological changes increases with growing population pressure. Thus, to assess where hydrological changes pose the biggest societal threat, we in combination analyse hydrological change and water crowding. Comprehensive results with the proportion of population in each crowding class that is more likely than not affected by severe changes in MAD, ND, Q10 or any combination of these three are shown in Fig. S3. For reasons of clarity we present the results for two aggregated classes of water crowding: high water crowding with >1000 p/fu and moderate to low water crowding with ≤1000 p/fu (Fig. 3). ¶
**Under the assumption of no climate change, as much as 2.99 billion people (33.3 % of total**

change if $\Delta T_{glob}$ would rise to 5 °C by 2100 (Fig. 3a). Out of these, 538 million (6.0 % of global population), 500 million (5.7 % of global population), and 640 million (7.1 % of global population) would more *more likely than not* be exposed to a severe decrease in mean water availability, a severe increase in droughts, and a severe increase in flood hazard, respectively (Fig. 3b-d). For 875 million people a transgression of thresholds for a mix of different aspects of severe hydrological change in more than half of the GCMs. A successful implementation of the Paris Agreement that would limit warming to 2 °C would dramatically reduce the number of people under *absolute water scarcity* and *more likely than not* exposed to sever hydrological change to 290 million (3.2 % of global population). With even more ambitious mitigation efforts sufficient to limit warming to 1.5° C warming could further reduce this number to as little as 116 million people (1.3 % of global population). For a failure of the Paris Agreement with temperature rising to 2.5° C (3° C) this number would rise to 543 (824) million people.

10 The remaining number of people exposed by severe hydrological change at 2° C warming as well as the implications of more ambitious mitigation efforts, or a failure of the Paris Agreement, differs greatly among world regions (Table 1, for countries assigned to each region see Fig. S4). About 63 % of the 290 million people who live under *absolute water scarcity* and are *more likely than not* exposed to severe hydrologic change at 2 °C warming, live in Latin America (LAM) and the Middle East and North Africa region (MEA), where they make up more than 12 % of the population in those regions. Another 28 % of the

15 290 million live in South Asia (SAS) and sub-Saharan Africa (SSA), but due to high population numbers in these regions their share remains below 2 %. The high share of population affected by *absolute water scarcity* and severe hydrological change in LAM and MEA is particularly worrying since a failure to overcome the obstacles associated with the implementation of appropriate demand management can have negative societal and economic consequences not only for these people but for the whole region. More ambitious mitigation efforts that keep warming below 1.5 °C would reduce the number of affected people

20 by more than half, to 6.5 % in MEA and 4.2 % in LAM. In all other regions, the share of affected population would drop below 1%.

Failure of the Paris Agreement would substantially increase exposure to severe hydrological change in many regions. In five out of ten regions, the number of people affected by *absolute water scarcity* and severe hydrological change at least doubles if the 2 °C target is exceeded by only 0.5 °C and reaches a share of (almost) 5 % of affected population in the region in SSA,

25 North America (NAM), and Europe (EUR). The strongest absolute increase (though not a doubling) in the number of affected people occurs in the MEA region, where more than one quarter of the population in that region would be affected at 2.5° C warming. Between 2.5 °C and 3 °C warming the increases in number of affected people is strongest in South Asia (SAS), SSA, North America (NAM), and EUR. At 4 °C warming, the share of affected population exceeds 10 % in 7 out of 10 regions, with MEA, Australia-New Zealand, SAS, SSA, and LAM being most strongly impacted. At 5 °C warming, the share of affected

30 population reaches 43.3 % in MEA and 35.0 % in ANZ, exceeds 20 % in SAS, SSA, and LAM, and exceeds 15 % in NAM, EUR, and East Asia (EAS). In Russia and Central Asia (RCA) and Southeast Asia (SEA) the share of affected people remains below 5 %, partly due to a low share of population under high water crowding and less severe hydrologic change.

Although numbers differ among population scenarios, the overall pattern of where and how much change occurs in the different regions is consistent across all SSP population scenarios. A comprehensive overview over population under high water crowding and affected by severe hydrologic change in different world regions for all population scenarios is given in Fig. S5.

**4 Discussion**

5    Our estimate that 26.8 % of global population today live under *absolute water scarcity* (>1000 p/fu), is well within the range of 21.0–27.5 % (average 24.7 %) reported by previous studies applying the WCI on river basin level (Gerten et al. 2013; Arnell and Lloyd-Hughes 2014; Kummu et al. 2016). Estimates of future SSP population living in river basins with >1000 p/fu under present-day climate conditions are given by Arnell & Lloyd-Hughes (2014), who estimate a range of 39.5–54.2 % of affected global population across different SSP scenarios. This is considerably higher than the range of our estimates of 31.5 – 44.9 %,
10   but due to the lack of other comparable studies, it is not clear whether these discrepancies are caused by the choice of the hydrological model or by the difference in scale (basin or grid cell) at which the WCI is calculated. However, using the same hydrological model as in our study, Gerten et al. (2013) estimate that 38.5 % of global population in the SRES A2r population scenario would live in river basins with >1000 p/fu under current climate conditions, which is close to our estimate of 41.0 % for the SSP3 scenario, to which the A2r scenario is comparable in terms of total population (12.3 billion compared to 12.6
15   billion in 2100). In contrast, the corresponding estimate from by Arnell & Lloyd-Hughes (2014) is as high as 54.2 % for the SSP3 scenario, which indicates that using LPJmL to assess water scarcity generally tends to result in lower estimates of future population affected by water scarcity.

A direct comparison of hydrological changes estimated here to previous studies is not straightforward due to the unique design of this study. Only few global studies have assessed climate change impacts on water resources as function of $\Delta T_{glob}$ (Gerten
20   et al. 2013; Schewe et al. 2014; Gosling and Arnell 2016), but they typically focus on changes in mean annual discharge and report changes in number of people affected by water scarcity. A relevant study for comparison is Schewe et al. (2014), which analyses changes in MAD obtained from an ensemble of ten global hydrological models (GHMs) forced by climate scenarios from five different GCMs. The overall pattern of changes in MAD simulated by LPJmL across 19 GCMs agrees well with results from Schewe et al. (2014), but exhibits a generally lower magnitude of changes (see Fig. S6 and Fig. 1 in Schewe et al.
25   (2014)). Thus, MAD changes simulated by LPJmL (both increases and decreases) tend to be smaller than simulated by most other GHMs. This becomes even more apparent when comparing the percentage of people affected by a 20 % decrease in MAD. For a $\Delta T_{glob}$ of 2.5 °C (equivalent to an additional warming of 1.9 °C relative to the control simulation) we estimate a median share of 8.6 % of affected global population across all GCMs. This is substantially lower than the median value of 13 % of affect population estimated for 2 °C additional warming by Schewe et al. (2014) and approximately represents their
30   lower end of the interquartile range. This can be attributed to the response of dynamic vegetation in LPJmL that is not included in most other GHMs (Schewe et al. 2014).

In summary, the global and regional estimates of population living under *absolute water scarcity* and being exposed severe hydrological change obtained from LPJmL are lower than from most other GHMs. Thus, the estimates of population affected by water scarcity and severe hydrological change presented in this paper should be regarded as conservative estimates.

Apart from these uncertainties in model projections, the results of a global study like ours are necessarily determined by simplifications and generalization in the data analysis. The most important generalization in this study are the choice of aspects of severe hydrological change and the corresponding critical thresholds. While not all selected aspects may be relevant in all cases (e.g., where supply is primarily fulfilled from groundwater), we believe that in the vast majority of cases they reflect important hydrological properties that are relevant from a freshwater resource perspective. The respective thresholds may also differ depending on hydrological and other local conditions, and using unique global values will always produce a number of false positives and false negatives. However, the selected thresholds are rather conservative, and thus are expected to produce more false negatives than false positives. Another aspect is the choice of the WCI to differentiate population groups in terms of adaptation challenges. This indicator is widely applied because it requires only data on mean water availability and population numbers, but it can account neither for hydrological aspects that limit the utilisation of water resources nor for actual per capita water requirements. Despite these shortcomings of the WCI, it gives a rough impression of the overall population pressure on water resources, which is linked to the challenges to adapt to severe hydrological change. Last but not least it is important to note that this study only addresses quantity aspects of freshwater resources and does not consider water quality.

**5 Conclusions**

Future freshwater supply will be affected by population growth and climate change, which are both subject to uncertainty and heterogeneous distribution patterns. Under all five SSP population projections considered here, a strong increase in the number of people living under *absolute water scarcity* in 2100 to 2.16 - 5.65 billion (31.5 - 44.9 % of global population) is projected, with higher global population resulting in higher absolute numbers but also larger proportions of global population being affected. Because of the importance of water demand management for coping with *absolute water scarcity*, which is more difficult to implement than supply management, these parts of population will face higher challenges for adaption to severe hydrological change that affects water supply.

If global warming would continue unabated to reach 5 °C above pre-industrial levels in 2100, 4.93 billion people (54.9 % of global population) in the SSP2 population scenario would *more likely than not* be exposed to severe hydrological change. Out of those, 2.14 billion people (23.9 % of global population) would already experience *absolute water scarcity* due to high population pressure on water resources, where adaptation to such changes is more challenging. With a successful implementation of the Paris Agreement limiting global warming to 2 °C, the number of people affected by severe hydrological change could be reduced to 615 million people (6.9 % of global population), of which 290 million (3.2 % of global population) would already experience *absolute water scarcity*. If temperature increase could be limited to 1.5 °C, the number of people

exposed to climate-driven water challenges could be further reduced to 195 million (2.2 %) and 116 million (1.3 %), respectively. However, only a partial failure of the Paris Agreement with temperature rising to 2.5° C would almost double the number of people *more likely than not* exposed to severe hydrological change, in total and among those already experiencing *absolute water scarcity*, compared to a 2° C warming.

5   Due to the heterogeneous spatial distribution of *absolute water scarcity* and severe hydrological change, the proportion of population exposed to sever hydrological change with increased adaption challenges reaches 12.0 % in the Latin America and 13.8 % in the Middle East and North Africa region even if global warming could be limited 2 °C by a successful implementation of the Paris Agreement. A failure to overcome the obstacles associated with the implementation of appropriate demand management can have negative societal and economic consequences not only for these people but for the whole region.

10  Thus, 2 °C mean global warming cannot be considered a safe limit of warming in these regions. More ambitious mitigation efforts that would keep warming at, or below, 1.5 °C could substantially reduce that risk by reducing the share of population exposed to severe hydrological change and with increased adaption challenges by more than half in these two regions and globally.

[Figure]

[Figure]

**Figure 1: Spatial pattern of water crowding in 2010 (a) and in 2100 for SSP2 population (b). Absolute (c) and percentage share of total population (d) in different water crowding classes from 1950 to 2010 and from 2011 to 2100 in five different SSP population scenarios under current water availability, i.e. assuming no climate change.**

[Figure]

[Figure]

**(a) Severe hydrological change (ΔMAD < −20% ∨ ΔND > 50% ∨ ΔQ10 > 30%)**

Affected population in SSP2 by 2100 (billion)

0   1   2   3   4   5   6   7   8   9

**(b) Severe decrease in mean water availability (ΔMAD < −20%)**

Affected population in SSP2 by 2100 (billion)

0   1   2   3   4   5   6   7   8   9

**(c) Severe increase in droughts (ΔND > 50%)**

Affected population in SSP2 by 2100 (billion)

0   1   2   3   4   5   6   7   8   9

**(d) Severe increase in flood hazard (ΔQ10 > 30%)**

Affected population in SSP2 by 2100 (billion)

0   1   2   3   4   5   6   7   8   9

■ 1.5K  ■ 2.0K  ■ 2.5K  ■ 3.0K  ■ 3.5K  ■ 4.0K  ■ 4.5K  ■ 5.0K  ■ no transgression

**(a) MAD**

0   1   2   3   4   5   6   7

**(c) Flood**

0   1   2   3   4   5   6   7

**Figure 2:** $\Delta T_{glob}$ at which severe hydrological changes occur in more than half of the GCMs (10 out of 19). Bars underneath the maps indicate population exposed to the respective severe changes for the SSP2 population scenario.

[Figure]

[Figure]

Figure 3: Fraction of SSP2 population in 2100 exposed to severe hydrological change at different levels of $\Delta T_{glob}$ (as shown in Fig. 2) divided over two water scarcity categories: population already experiencing absolute water scarcity (>1000 p/fu) in the absence of climate change and rest of population (≤1000 p/fu). The total number of people in each class is given on the y-axis, and the fraction of people exposed to sever hydrological change in each class is given on the x-axis. Color scale for $\Delta T_{glob}$ same as in Fig. 2.

**Table 1: Number of people in 2100 for the SSP2 population scenario** that would experience absolute water scarcity (>1000 p/fu) under present-day climate conditions and be *more likely than not* exposed to severe hydrological change at different levels of $\Delta T_{glob}$ in different world regions (population in million, percentage of population in region in brackets). Regions are: MEA (Middle East and North Africa), ANZ (Australia and New Zealand), SAS (South Asia), SSA (Sub-Saharan Africa), LAM (Latin America), NAM (USA and Canada), EUR (Europe, excluding Russia), EAS (East Asia), RCA (Russia and Central Asia), SEA (Southeast Asia).

| | Total Population | Population > 1000 p/fu | Population with > 1000 p/fu and exposed to severe hydrologic change | | | | | |
|---|---|---|---|---|---|---|---|---|
| | | | 1.5 °C | 2.0 °C | 2.5 °C | 3.0 °C | 4.0 °C | 5.0 °C |
| MEA | 740 | 416 (56.2%) | 48.0 (6.5%) | 101.9 (13.8%) | 193.2 (26.1%) | 222.4 (30.0%) | 270.9 (36.6%) | 320.4 (43.3%) |
| ANZ | 51 | 27 (52.2%) | 0.0 (0.0%) | 0.6 (1.2%) | 1.5 (2.9%) | 3.7 (7.3%) | 9.6 (18.8%) | 17.8 (35.0%) |
| SAS | 2282 | 1005 (44.1%) | 17.1 (0.7%) | 38.7 (1.7%) | 80.7 (3.5%) | 200.8 (8.8%) | 461.8 (20.2%) | 651.2 (28.5%) |
| SSA | 2395 | 890 (37.2%) | 16.8 (0.7%) | 42.8 (1.8%) | 113.2 (4.7%) | 191.0 (8.0%) | 371.5 (15.5%) | 575.9 (24.0%) |
| LAM | 662 | 230 (34.7%) | 27.7 (4.2%) | 79.8 (12.0%) | 87.5 (13.2%) | 101.0 (15.3%) | 131.3 (19.8%) | 175.2 (26.5%) |
| NAM | 510 | 166 (32.6%) | 4.2 (0.8%) | 6.4 (1.2%) | 26.9 (5.3%) | 39.3 (7.7%) | 58.7 (11.5%) | 86.8 (17.0%) |
| EUR | 579 | 161 (27.9%) | 0.4 (0.1%) | 14.6 (2.5%) | 31.0 (5.4%) | 42.8 (7.4%) | 70.0 (12.1%) | 90.6 (15.7%) |
| EAS | 913 | 253 (27.7%) | 0.1 (0.0%) | 2.4 (0.3%) | 4.4 (0.5%) | 16.5 (1.8%) | 61.7 (6.8%) | 163.3 (17.9%) |
| RCA | 198 | 44 (22.1%) | 1.9 (1.0%) | 3.1 (1.6%) | 5.0 (2.5%) | 6.9 (3.5%) | 9.7 (4.9%) | 11.1 (5.6%) |
| SEA | 642 | 106 (16.5%) | 0.0 (0.0%) | 0.0 (0.0%) | 0.0 (0.0%) | 0.0 (0.0%) | 12.2 (1.9%) | 48.7 (7.6%) |
| World | 8971 | 3298 (36.8%) | 116 (1.3%) | 290 (3.2%) | 543 (6.1%) | 824 (9.2%) | 1457 (16.2%) | 2141 (23.9%) |